# 4G cloning: rapid gene assembly for expression of multisubunit protein complexes in diverse hosts

Michael Taschner, Joe Bradley Dickinson, Florian Roisné-Hamelin, Stephan Gruber

**Multisubunit protein complexes are central to many cellular processes, and studying their activities and structures in vitro requires reconstitution via recombinant expression and purification. Obtaining targets at sufficient purity and scale typically involves screening several protein variants and expression hosts. Existing cloning strategies enable co-expression but are often time-consuming, labor-intensive, and host-specific, or involve error-prone steps. We present a novel vector set and assembly strategy to overcome these limitations, enabling expression construct generation for multisubunit complexes in a single step. This modular system can be extended to additional hosts or include new tags. We demonstrate its utility by constructing expression vectors for structural maintenance of chromosomes complexes in various hosts, streamlining workflows, and improving productivity.**


## Introduction

Expression and purification of recombinant proteins is an integral part of many areas in molecular biology and biochemistry. Obtaining sufficient quantities of such samples at the desired level of purity frequently requires significant efforts in screening numerous expression constructs before finding one that gives satisfactory results.

Two decisions are crucial before starting to express and purify a new target. First, one has to select a suitable expression host (1). Bacterial expression in *Escherichia coli* is usually the first choice because of the low cost, fast progress, minimal need for non-standard laboratory equipment, and ease of upscaling. Eukaryotic proteins, however, often require more complex machinery for folding and/or post-translational modification and thus benefit from eukaryotic expression hosts. Second, an appropriate affinity tag needs to be found that allows for efficient purification of the target in a minimal number of steps without negatively impacting folding and function of the tagged protein.

For multisubunit complexes, the individual subunits can be expressed from separate vectors, which are co-delivered into host cells, but this becomes inconvenient and unreliable for larger assemblies (>3 subunits). A better alternative is to produce multiple proteins from the same vector, with each subunit being expressed from its own gene expression cassette (GEC) with appropriate regulatory sequences. In recent years, several systems have been developed to facilitate the production of such multi-GEC plasmids for bacteria, insect cells, or mammalian cells (see, e.g., references (2, 3, 4, 5, 6)). They are, however, host-specific and not easily cross-compatible. Moreover, cloning of constructs involves multistep procedures that initially require the generation of single expression constructs (tagged or untagged), which are subsequently combined into larger assemblies. Modification to the expression strategies at a later time, such as changing tags or switching expression hosts, is often not straightforward and requires extensive new efforts. As an alternative to multi-GEC expression, several ORFs can be expressed from a single cassette by connecting them via internal binding sites for the ribosome ("RBS" for bacteria; "IRES" for eukaryotic cells) to create polycistronic constructs, or by a peptide linker of variable lengths to produce fusion proteins.

Traditional cloning methods (by restriction and ligation) are often labor-intensive, especially for multisubunit expression vectors, as they typically require extensive handling and purification of DNA fragments (PCR purification, agarose gel extraction, etc.), and resequencing of intermediates and final products if PCR techniques are employed. More recent developments offered convenient alternatives to these traditional approaches.

Golden Gate cloning uses Type IIS restriction enzymes, which cleave DNA outside their non-palindromic recognition sequence, to assemble one or more sequences provided in "donor" vectors in the correct order into "acceptor" vectors (7, 8). Donor and acceptor vectors differ in the placement and orientation of the Type IIS sites. Cloning is carried out in a "one-pot restriction/ligation" reaction in which the desired circular plasmid (devoid of any recognition sites for the particular enzyme) accumulates. All fragments must have unique "sticky ends" created by the Type IIS enzyme, and any internal restriction sites in target sequences have to be removed in a process termed "sequence domestication." Because of the simple cut-and-paste mechanism without any PCR intermediate, the product does not need to be resequenced if the sequences of the acceptors and donors have already been adequately verified.

Department of Fundamental Microbiology (DMF), Faculty of Biology and Medicine (FBM), University of Lausanne (UNIL), Lausanne, Switzerland

Correspondence: michael.taschner@unil.ch; stephan.gruber@unil.ch

Gibson assembly (9) allows the construction of a target sequence from several linear fragments into a linearized vector in a short isothermal reaction, using short (15–25 bp) stretches of homology at the desired junctions. Fragments are often generated by PCR, and the product thus requires sequence verification. Even if the fragments are created by error-free means, the assembly process involves the action of a DNA polymerase 100–200 bp around fragment junctions (9), so at least these areas need to be sequence-verified if they contain critical elements.

Golden Gate assembly (e.g., references (4, 6)), Gibson assembly (e.g., reference (3)), or a combination thereof (10) has been successfully employed to create multi-GEC plasmids, but they either (i) use a stepwise cloning approach over multiple days, which often requires tedious DNA fragment purification and handling steps, (ii) involve intermediate PCR steps and thus require the large final product to be sequence-verified, or (iii) are specifically designed to target only one particular expression host.

Here, we present a new cloning strategy, which we term "**G**olden **G**ate–**g**uided **G**ibson assembly" or "4G cloning" for short, to produce complex expression vectors for several expression hosts. It solves the aforementioned problems and allows quick and reliable assembly of many construct variations in parallel. We provide a set of vectors that can easily be expanded to include new tags or regulatory sequences. For expression in insects and mammalian cells, the vectors are fully compatible with the recently described biGBac method (3). Our system allowed us to create several bacterial expression constructs for screening of the hexameric Smc5/6 complex from *Saccharomyces cerevisiae*, as well as similar constructs for the *Schizosaccharomyces pombe* and *Homo sapiens* Smc5/6 hexamers for expression in insect and mammalian cells, respectively. Moreover, the strategy has been successfully applied to produce the three-subunit Wadjet structural maintenance of chromosomes (SMC) complex for biochemical reconstitution and structural analysis. Lastly, although our work presented here is focused on the production of vectors for protein expression, the same assembly pipeline (4G cloning) can be applied to other areas requiring quick modular assembly of repeating building blocks.

# Results

### Assembly of multi-GEC expression plasmids using *G*olden *G*ate–*g*uided *G*ibson assembly

Here, we devised a cloning strategy based on Gibson assembly of Golden Gate–customized GECs, producing final multigene expression vectors from sequence-verified elements in a single cloning step (Fig 1) ("**G**olden **G**ate–**g**uided **G**ibson assembly" or 4G cloning). Briefly, Golden Gate assembly of compatible DNA fragments first produces a linear product, the GEC. Multiple GECs with appropriate terminal overhangs are then directly mixed and inserted into an acceptor vector in a defined order by Gibson assembly. We prepared a DNA-element library (Table 1), with which production of multi-GEC expression plasmids can be carried out in a single day with minimal hands-on time. First, Golden Gate reactions are set up individually for each GEC including donors for all desired elements (Fig 1, top half), with complete flexibility regarding the inclusion of a tag, host-specific regulatory sequences, and position of the GEC in the final construct. Without further amplification or purification, the products of these reactions are then combined and mixed with a linearized acceptor vector to produce the desired plasmid by Gibson assembly on the same day (Fig 1, bottom half). Upon transformation into *E. coli*, positive clones can be identified and then used directly for expression screening without further sequencing, because GECs are assembled from sequence-verified parts by Golden Gate assembly, and any mutation introduced during Gibson assembly is limited to neutral spacer sequences surrounding the junction sites.

We started by creating the building blocks, here designated as "elements," for GEC assembly in a set of Golden Gate donor plasmids. Examples are shown schematically in Fig S1A, and a more detailed view is provided in Fig S1D–F. Each donor plasmid contains one or two elements (such as an ORF-element, a TAG-element, promoter-elements and terminator-elements, or Gibson-elements) (Fig S1 and Table 1), which include appropriate flanks for subsequent Golden Gate assembly (see the Materials and Methods section). The flanks harbor recognition sites for the enzyme *BsaI*, cleavage of which creates unique 4-bp "sticky ends" (numbers/letters in white circles in all figures). The cloning of these elements into donor plasmids by PCR and Gibson assembly (Fig S1B) also allows for sequence verification and for concomitant sequence domestication, that is, the removal of any internal *BsaI* sites in target sequences (Fig S1C). All donors have a backbone containing a chloramphenicol resistance marker ($Cam^R$) and a conditional origin of replication (R6Kγ) only functional in bacteria expressing the *pir* gene (Fig 1, top right).

Donors for ORF-elements need to be created and sequence-verified by the user for each new expression target, that is, each subunit of a protein complex of interest. To ensure that ORFs can be tagged at their C-terminus, it is important to remove the native STOP codon when cloning ORF-elements (see details in the Materials and Methods section and Fig S1E). When such an ORF is cloned into the expression vector without a tag, the resulting protein will have its native N-terminus but an additional glycine residue at the C-terminus because of the addition of a GGA triplet coming from the *BsaI* overhang. Inclusion of the native STOP codon into the reverse primer used for ORF amplification will yield a donor for the expression of a protein containing the native C-terminus. This will be relevant in cases where protein function depends on an unmodified C-terminus or where C-terminal tagging is not desired.

The remaining set of vectors (Table 1) is available from Addgene. Donors for N- and C-terminal TAG-elements carry inserts that can be fused with the 5′- or the 3′-end of an ORF-element, respectively. We created such TAG-elements for many commonly used protein purification tags, with several of them including protease recognition sites for optional tag removal (TEV or 3C; Table 1), but new donors for any additional tag-elements can be created using the same strategy (Fig S1B and C). Upstream and downstream sequences are provided by promoter-elements and terminator-elements, generally combined in "P+T"-donors. We created such P+T combinations for expression in bacteria (T7 promoter; T7 terminator), in insect cells (polyhedrin promoter; SV40-polyA), and in mammalian cells (CMV promoter; bGH-polyA). Importantly, the P+T-

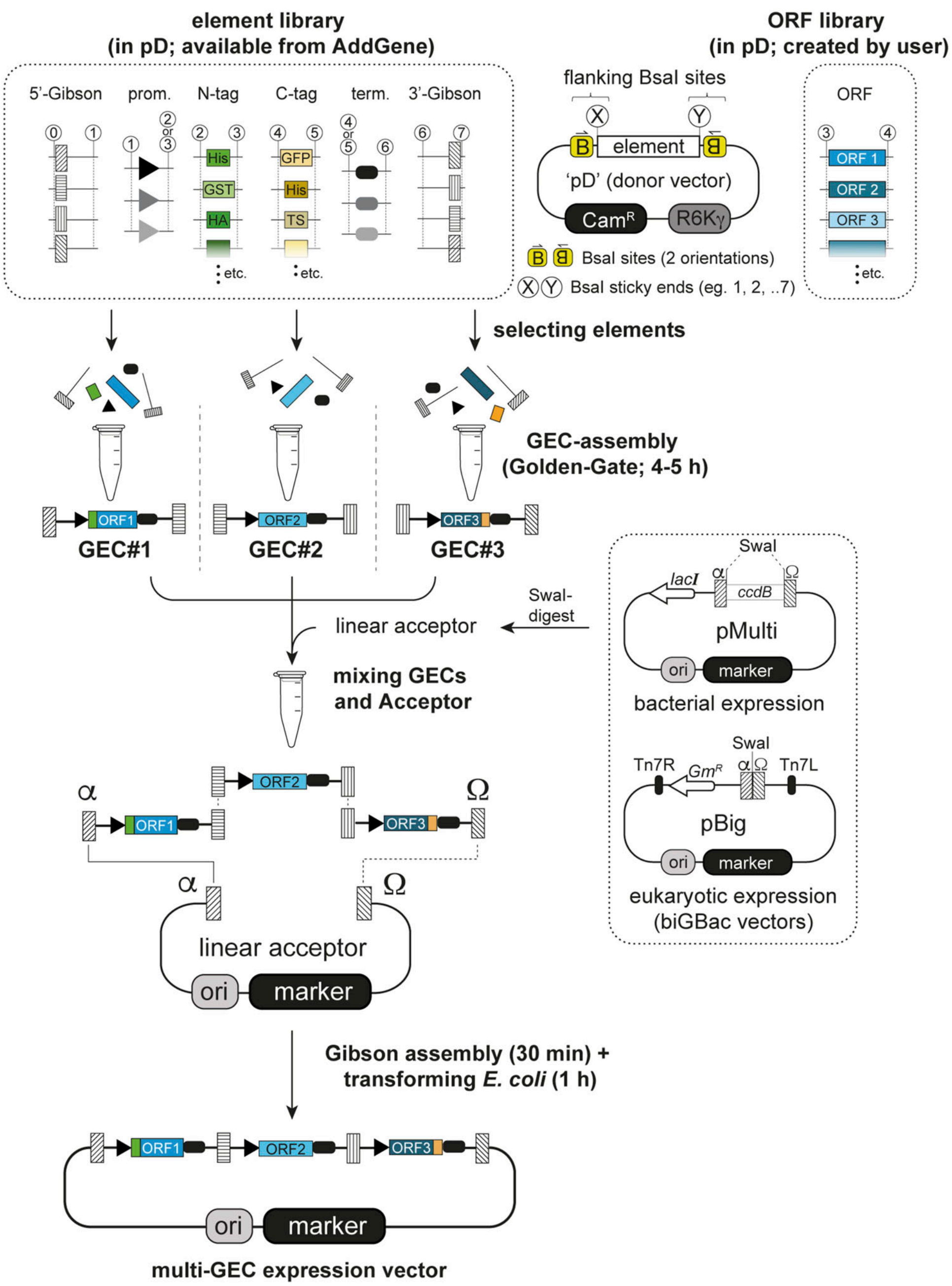


**Figure 1. Overview of the "Golden Gate–guided Gibson" (4G cloning) procedure.**
In the first step, individual linear gene expression cassettes (GECs) are assembled from individual smaller elements by Golden Gate cloning. These elements are excised from individual circular donor vectors ("pD"; see generic representation next to the element library box), giving specific sticky ends depending on the element type (numbers in white circles). Multiple GECs with compatible homology ends from such reactions are then directly inserted into dedicated linearized acceptor vectors by Gibson assembly without further isolation/purification. As a result, a multi-GEC expression plasmid can be created in a single day from prepared donors.

donors each exist in three forms to allow the untagged expression or addition of an N- or C-terminal tag-element (Fig S1A and E, and Table 1). Lastly, donors for 5′- and 3′-Gibson-elements provide sequences that attach upstream of the promoter-element and downstream of the terminator-element, respectively (Fig S1A and F). They add terminal homology regions of around 20 bp for Gibson assembly, which are separated from the internal elements of the GEC by 200- to 300-bp-long spacers (to avoid sequence alterations in critical parts during Gibson assembly).

We also built acceptor vectors for the final multi-GEC assemblies allowing for expression in bacteria. Three pMulti-plasmids differing in their antibiotic resistance marker (ampicillin, kanamycin, or streptomycin; see the box on the right in Fig 1) contain a *ccdB* suicide cassette to avoid vector background during cloning, and thus are maintained in appropriate *ccdB*-resistant strains. Digestion with the enzyme *SwaI* removes the suicide cassette and exposes homology regions (α and Ω) used for GEC insertion by Gibson assembly. For creating a multi-GEC plasmid for insect and mammalian expression (see below), we use plasmids pBig1a-pBig1e from the biGBac system (plasmid kit #1000000088; available from Addgene), which are linearized in the same way. These vectors are compatible because our α and Ω overhangs are identical to those described originally for the biGBac system (3).

Correct formation of the desired multi-GEC vectors depends on the complete assembly and subsequent joining of linear GECs. We reasoned that the efficiency would decrease with an increasing number of GECs and set out to test this. We determined the efficiencies of obtaining clones containing either two, three, or four GECs by counting the number of colonies after transformation of the 4G cloning reaction (see Fig S1G for an overview of the assembly process). As expected, the total number of clones decreased progressively from around 1,000 for two GECs to around 100 for four GECs (Fig 2A and B, top), whereas in the absence of any GEC, the vector alone gave neglectable background. We then randomly picked five colonies for each plate, isolated the candidate plasmid, and analyzed it by an *EcoRV* digest. Although the accuracy of assembly decreased with an increasing number of GECs, around 50% of the picked clones with four GECs were still correct (Fig 2A and B, bottom). As expected, whole-plasmid sequencing of selected constructs showed the absence of unwanted mutations or assembly errors. Of note, to allow simultaneous assembly of even more GECs into a single vector, additional Gibson-element donors will have to be designed. Moreover, we would like to underscore that high-quality miniprep DNA (devoid of nuclease contaminations) and fresh enzyme stocks are important for efficient assembly. We usually store small aliquots of the enzyme at −70°C.

Because of the quick and easy assembly process, 4G cloning allows to create many versions of expression plasmids in parallel. For example, if the best position for a purification tag is unknown, three GEC versions for each subunit (untagged, with an N-terminal tag, or with a C-terminal tag) can be prepared and then combined in a way to screen all possibilities without significantly increasing the workload (see below). Also, if a certain expression host (like *E. coli*) turns out to be a poor choice for production of a given target, similar vectors for a different host can be created with minimal adjustments (i.e., usage of different promoter- and terminator-elements and acceptor vectors). It should, however, be noted that using the same ORF-donors for production of vectors for pro- and eukaryotic hosts may lead to problems with codon usage. If ORF sequences are synthesized, one might want to choose codons, which are suitable for several hosts. We have recently reported the successful application of this cloning strategy for the expression of two distinct multisubunit protein complexes in *E. coli* (11, 12, 13, 14). Here, we briefly describe the initial creation and further development of these expression tools based on a challenging target complex.

## Expression of the hexameric Smc5/6 complex from budding yeast in *E. coli*

SMC complexes regulate essential aspects of chromosome structure and segregation in all domains of life, and share a common architecture containing two SMC proteins and several non-SMC components (15, 16). Eukaryotes contain three distinct SMC complexes called cohesin, condensin, and Smc5/6, with prominent roles in sister chromatid cohesion, chromosome condensation, and DNA repair, respectively. Whereas recombinant production of active SMC complexes has been achieved using eukaryotic expression hosts, the obtained yields and purity are limited, particularly for the hexameric yeast Smc5/6 complexes. At its core, this complex contains the SMC proteins Smc5 and Smc6, as well as four non–SMC-elements (Nse1-Nse4) (17). Evidence in the literature indicates that several of these proteins from various species can be produced as smaller subcomplexes in *E. coli* (18, 19).

We set out to test production of the hexameric holocomplex from budding yeast in *E. coli* using 4G cloning; we prepared the six necessary ORF-elements by PCR amplification of the respective coding sequences from the yeast genome, followed by Gibson assembly into the linear donor vector according to Fig S1B and C. Internal *BsaI* sites for Nse2, Smc5, and Smc6 were mutated as shown in Fig S1C. Having no prior knowledge about the optimal type and position of affinity tag for this complex, we decided to test all possible 12 positions for a single Twin-Strep-tag-element in an unbiased way. An overview of our cloning procedure is shown in Figs 3A and S2. We first assembled three GECs with T7 promoter-element and T7 terminator-element for each subunit (for untagged and N- or C-terminally Twin-Strep–tagged versions), giving a total of 18 GECs. Upstream and downstream homology-elements were chosen to create vectors with three GECs (*Nse4*/*Nse3*/*Nse1* and *Smc6*/*Smc5*/*Nse2*). Golden Gate products were mixed in several combinations with linearized pMulti-plasmids to give 14 final expression vectors. The *Nse4*/*Nse3*/*Nse1* combinations were inserted into pMulti_K (with $Kan^R$), whereas the Smc6/Smc5/Nse2 combinations were inserted into pMulti_A (with $Amp^R$). Seven plasmids for each GEC combination were created, one with all subunits untagged and six for all possible tag positions (see Fig S2 for a detailed view). Importantly, all these vectors were created in parallel in a single day. *E. coli* cells (the Rosetta [DE3] cell line was chosen because the coding sequences were not codon-optimized for bacteria) were then co-transformed with 13 combinations of these plasmids (12 for the possible tag positions and one with all untagged subunits as a negative control for purification) for small-scale test expressions and Strep-Tactin pulldowns. The result of this experiment is shown in Fig 3B. The minimal background was observed after Strep-Tactin pulldowns using extracts of cells expressing the untagged complex (lane 1). Tagging of Nse1, Nse2, Nse3, Nse4, or Smc5 gave similar results, regardless of whether the Twin-Strep-tag was added to the N- or the C-terminal end of the respective protein. These pulldowns yielded protein amounts clearly detectable by Coomassie staining, but the band for Smc6 was always clearly weaker than the bands for the other subunits (lanes 2–11), indicating poor subunit stoichiometry. When the tag was placed at the N-terminus of Smc6, the amount of protein obtained in the pulldowns was minimal (lane 12), indicating that Smc6 does not tolerate this tag position. Whether this is due to the lack of the expression of this modified subunit or inaccessibility of the tag is currently not known. Moving the tag to the Smc6 C-terminus, however, yielded material after pulldown. Although the overall yield here (lane 13) was lower than with tags on any of the other subunits (lanes 2–11), the resulting complex appeared to have balanced subunit stoichiometry.

## Tag screening for the JetABC complex from *E. coli*

Bacteria have evolved sophisticated defense systems to fight incoming phages and so-called "selfish" DNA-elements. One of them, "Wadjet"/JetABCD (also called MksBEFG/EptABCD), restricts plasmid transmission. It encodes for a sensor component, JetABC, which is an SMC complex resembling the bacterial condensin MukBEF, associated with the effector JetD, a TOPRIM domain–containing

**Table 1. Plasmids for 4G cloning.**

| | Plasmid name | Resistance | Strain | Description | Addgene # |
|---|---|---|---|---|---|
| Acceptors | pMulti_A_ccdB | Amp | ccdB Surv. | GEC acceptor for *E. coli* expression, ampicillin resistance | 223851 |
| | pMulti_K_ccdB | Kan | ccdB Surv. | GEC acceptor for *E. coli* expression, kanamycin resistance | 223852 |
| | pMulti_S_ccdB | Strep | ccdB Surv. | GEC acceptor for *E. coli* expression, streptomycin resistance | 223853 |
| Promoter/terminator-donors | pD(P+T)_T7 | Cm | pir | Promoter/terminator for *E. coli* expression, no tag | 223854 |
| | pD(P+T)_T7N | Cm | pir | Promoter/terminator for *E. coli* expression, N-term. tag | 223855 |
| | pD(P+T)_T7C | Cm | pir | Promoter/terminator for *E. coli* expression, C-term. tag | 223856 |
| | pD(P+T)_ph | Cm | pir | Promoter/terminator for insect expression, no tag | 223857 |
| | pD(P+T)_phN | Cm | pir | Promoter/terminator for insect expression, N-term. tag | 223858 |
| | pD(P+T)_phC | Cm | pir | Promoter/terminator for insect expression, C-term. tag | 223859 |
| | pD(P+T)_CMV | Cm | pir | Promoter/terminator for mammalian expression, no tag | 223860 |
| | pD(P+T)_CMVN | Cm | pir | Promoter/terminator for mammalian expression, N-term. tag | 223861 |
| | pD(P+T)_CMVC | Cm | pir | Promoter/terminator for mammalian expression, C-term. tag | 223862 |
| Gibson overhang donors | pD(Gib)_5′alpha | Cm | pir | Gibson overhang, adds alpha-sequence to the 5′-end | 223863 |
| | pD(Gib)_5′beta | Cm | pir | Gibson overhang, adds beta-sequence to the 5′-end | 223864 |
| | pD(Gib)_5′gamma | Cm | pir | Gibson overhang, adds gamma-sequence to the 5′-end | 223865 |
| | pD(Gib)_5′delta | Cm | pir | Gibson overhang, adds delta-sequence to the 5′-end | 223866 |
| | pD(Gib)_3′beta | Cm | pir | Gibson overhang, adds beta-sequence to the 3′-end | 223867 |
| | pD(Gib)_3′gamma | Cm | pir | Gibson overhang, adds gamma-sequence to the 3′-end | 223868 |
| | pD(Gib)_3′delta | Cm | pir | Gibson overhang, adds delta-sequence to the 3′-end | 223869 |
| | pD(Gib)_3′omega | Cm | pir | Gibson overhang, adds omega-sequence to the 3′-end | 223870 |
| N-terminal tag donors | pD(Nt)_His6 | Cm | pir | N-terminal hexa-histidine, non-cleavable | 223871 |
| | pD(Nt)_His6-TEV | Cm | pir | N-terminal hexa-histidine, TEV-cleavable | 223872 |
| | pD(Nt)_His6-3C | Cm | pir | N-terminal hexa-histidine, 3C-cleavable | 223873 |
| | pD(Nt)_His6-FLAG3 | Cm | pir | N-terminal hexa-histidine + triple-FLAG, non-cleavable | 223874 |
| | pD(Nt)_His6-FLAG3-TEV | Cm | pir | N-terminal hexa-histidine + triple-FLAG, TEV-cleavable | 223875 |
| | pD(Nt)_His6-HA2-TEV | Cm | pir | N-terminal hexa-histidine + double-HA, TEV-cleavable | 223876 |
| | pD(Nt)_His6-MBP-TEV | Cm | pir | N-terminal hexa-histidine + MBP, TEV-cleavable | 223877 |
| | pD(Nt)_His6-GST-TEV | Cm | pir | N-terminal hexa-histidine + GST, TEV-cleavable | 223878 |
| | pD(Nt)_His6-SUMO | Cm | pir | N-terminal hexa-histidine + SUMO, Senp2-cleavable | 223879 |
| | pD(Nt)_GFP | Cm | pir | N-terminal GFP, non-cleavable | 223880 |
| | pD(Nt)_GFP-TEV | Cm | pir | N-terminal GFP, TEV-cleavable | 223881 |
| | pD(Nt)_GFP-3C | Cm | pir | N-terminal GFP, 3C-cleavable | 223882 |
| | pD(Nt)_TS-3C | Cm | pir | N-terminal Twin-Strep, 3C-cleavable | 223883 |
| | pD(Nt)_His10-TS-3C | Cm | pir | N-terminal deca-histidine + Twin-Strep, 3C-cleavable | 223884 |
| | pD(Nt)_HALO | Cm | pir | N-terminal HALO, non-cleavable | 223885 |

**Table 1. Continued**

| | Plasmid name | Resistance | Strain | Description | Addgene # |
|---|---|---|---|---|---|
| C-terminal tag donors | pD(Ct)_His8 | Cm | pir | C-terminal octa-histidine, non-cleavable | 223886 |
| | pD(Ct)_3C-His8 | Cm | pir | C-terminal octa-histidine, 3C-cleavable | 223887 |
| | pD(Ct)_3C-GFP | Cm | pir | C-terminal GFP, 3C-cleavable | 223888 |
| | pD(Ct)_3C-Venus-His8 | Cm | pir | C-terminal octa-histidine + Venus, 3C-cleavable | 223889 |
| | pD(Ct)_CPD-His10 | Cm | pir | C-terminal deca-histidine + CPD, phytate-cleavable | 223890 |
| | pD(Ct)_TS | Cm | pir | C-terminal Twin-Strep, non-cleavable | 223891 |
| | pD(Ct)_3C-TS | Cm | pir | C-terminal Twin-Strep, 3C-cleavable | 223892 |
| | pD(Ct)_3C-TS-His10 | Cm | pir | C-terminal deca-histidine + Twin-Strep, 3C-cleavable | 223893 |
| | pD(Ct)_Avi-3C-TS | Cm | pir | C-terminal AviTag + Twin-Strep, partially 3C-cleavable | 223894 |
| | pD(Ct)_FLAG | Cm | pir | C-terminal FLAG, non-cleavable | 223895 |
| | pD(Ct)_FLAG3 | Cm | pir | C-terminal triple-FLAG, non-cleavable | 223896 |
| | pD(Ct)_MYC9 | Cm | pir | C-terminal nona-myc, non-cleavable | 223897 |
| | pD(Ct)_PK6 | Cm | pir | C-terminal hexa-V5, non-cleavable | 223898 |
| | pD(Ct)_HALO | Cm | pir | C-terminal HALO, non-cleavable | 223899 |

All plasmids are available from Addgene, individually or as a kit (Addgene ID 1000000252). Amp, ampicillin; Kan, kanamycin; Strep, streptomycin; MBP, maltose-binding protein; GST, glutathione S-transferase; GFP, green fluorescent protein; CPD, cysteine protease domain.

nuclease (12, 20, 21). JetABC forms a dimer of motor units that identifies smaller circular DNAs by loop extrusion that are then cleaved by the JetD nuclease (12, 14, 20, 22 *Preprint*). We previously expressed the JetABC complex with an N-terminal "His-Twin-Strep-3C"-tag on the N-terminus of the kleisin subunit JetA (12), after screening for the best tag position using a strategy similar to the one described for Smc5/6. To screen other tags on the same subunit for specific applications (i.e., complex labeling) or for increased yield, we generated eight expression constructs in parallel by 4G cloning (Fig 3C). GECs for *JetC, JetB, and JetA* were created, with JetA being left either untagged or receiving four N- or three C-terminal tag combinations that all contain a His-tag for direct comparison by $Ni^{2+}$-NTA pulldown. The cassettes were inserted into the pMulti_A vector, and *E. coli* were transformed for expression tests. The standard BL21(DE3)Gold strain was chosen here because the coding sequences of the *E. coli* JetABC complex were naturally suitable for expression in bacteria. Bands for all three subunits (together with additional bands representing contaminants and/or degradation products) were observed even after pulldown of the untagged complex (Fig 3D, lane 1), consistent with a previous report that the JetC subunits bind to the His-affinity resin in a non-specific manner (22 *Preprint*). The yield here was, however, clearly lower than for the other constructs containing a tag. Among those, some produced more proteins than others (judging from the intensity of the Coomassie-stained bands), with the previously described construct (Fig 3D, lane 2) being among those. Large-scale expression using this plasmid yielded a properly assembled complex (see below).

### 4G cloning for expression in eukaryotic hosts

The above results demonstrate that quick cloning of multisubunit assemblies for expression in *E. coli* is feasible with 4G cloning, even for challenging targets like eukaryotic SMC complexes. However, we failed to express and purify Smc5/6 complexes from *S. pombe* and *H. sapiens* in *E. coli*, apparently because of the limited expression of the SMC subunits. To solve this issue, we shifted our attention to expression in eukaryotic hosts (insect and mammalian cells). We created promoter- and terminator-elements for expression in these hosts (polyhedrin promoter/SV40 polyA or CMV promoter/bGH-polyA, respectively) and performed 4G cloning with the respective ORF-elements cloned for *S. pombe* and *H. sapiens* Smc5/6 subunits. The combinations *Smc6/Smc5/Nse2* and *Nse4/Nse3/Nse1* were cloned into pBig1a and pBig1b, respectively, then combined into pBig2 according to the published procedure (3), and a similar tag-screening procedure as described for the budding yeast complex was carried out (Fig S3). The *S. pombe* subunits were put under the control of insect cell promoters, and the final vectors were used to produce baculoviruses, which were tested for protein expression in a small scale. The results shown in Fig 4A show that stoichiometric complexes can be obtained in a single purification step when tagging the SMC subunits, regardless of whether the tag is on the N- or the C-terminus. Tagging any of the smaller Nse subunits consistently yields excess of the small subunits, either of the Nse4/3/1 subcomplex or of Nse2.

For the expression of the human complex, we put the subunits under control of a CMV promoter and transfected the vectors resulting from the cloning procedure in Fig S3 into HEK293S cells (10-ml scale). Mammalian cell expression proved to be the most robust system, with the least variation regarding the placement of the affinity tag (Fig 4B). Even though a slight excess of the Nse4/3/1 complex can be noticed in the gel when one of these subunits is tagged, this would likely not pose any problems because this excess of the smaller subunits/subcomplex can be removed by size-exclusion chromatography. Our cloning strategy is thus easily adaptable to eukaryotic expression hosts and allows to

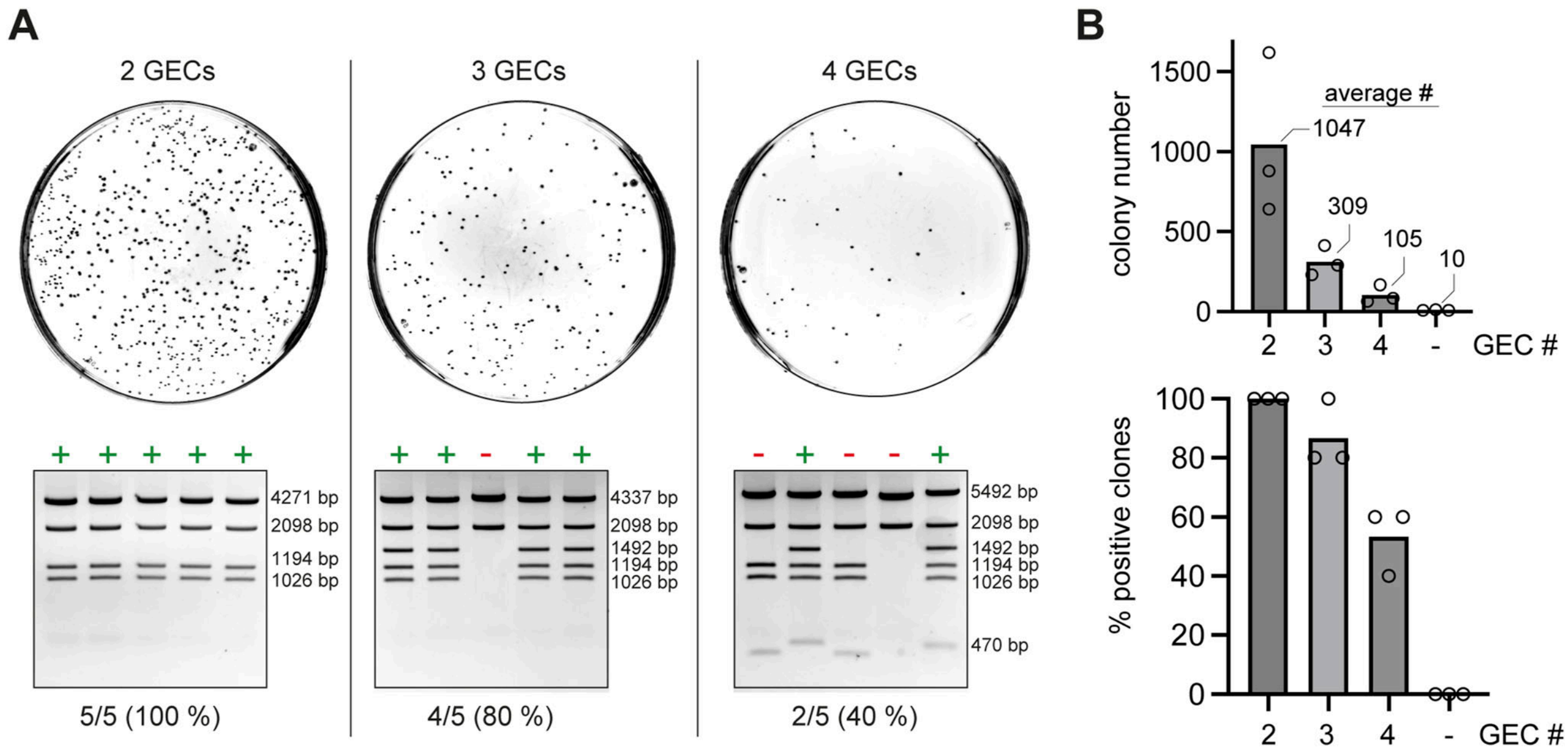


**Figure 2. Test of assembly efficiencies with two, three, or four GECs (using *S. cerevisiae* Nse1, Nse2, Nse3, and Nse4 ORF-elements as denoted in Fig S1G).**
**(A)** Competent *E. coli* Rosetta (DE3) cells (Novagen) were transformed with 25% of the final Gibson assembly reaction (2.5 out of 10 $\mu$l), and the total number of colonies was estimated (top). Plasmids were isolated from five randomly chosen clones and tested by *EcoRV* digest (bottom). **(B)** Quantification of obtained colony numbers and percentage of positive clones from three independent experiments. A control was included, which contained only the linear pMulti-acceptor without any GEC ("- GECs"). This control showed a very low number of background colonies. For colony counting, we used VisionWorks software (Analytik Jena), and manually inspected and corrected the count.
Source data are available for this figure.

perform unbiased screens for ideal expression constructs in these systems.

### Large-scale purification from bacteria and mammalian cells using selected constructs

The results presented in Figs 3 and 4 show protein complexes, which can be readily pulled down from lysates of small-scale cultures. This does not guarantee proper folding of the resulting material, which could be aggregated. To rule out this possibility, we went on to purify three of the four presented examples in a larger scale. For the Smc5/6 hexameric complex from *S. cerevisiae*, we chose a construct with a C-terminal Twin-Strep-tag on Smc6 (lane 13 in Fig 3B) as this version clearly gave material with appropriate subunit stoichiometry. Purification from 1 liter of bacterial culture using a three-step procedure clearly showed a dominant soluble peak of material after the final gel filtration step containing all six subunits (Fig 5A).

For the JetABC complex, we chose the construct with the N-terminal His-Twin-Strep-3C-tag on the JetA subunit. Because of the "stickiness" of the JetC to Ni-NTA resin (as observed in Fig 2D, lane 1), we used the Twin-Strep-tag for purification. From a 1-liter bacterial culture, a simple two-step purification procedure (Strep-Tactin-HP and size-exclusion chromatography) yielded again a large peak of soluble material containing all JetABC components (Fig 5B).

Lastly, we transfected 1 liter of HEK293 cell culture with an expression plasmid for the human Smc5/6 complex containing a C-terminal Twin-Strep-tag on the Nse4 subunit. After lysis, we followed a two-step purification procedure as for JetABC to obtain material containing all subunits, which was clearly not aggregated (Fig 5C). Mass spectrometric analysis also confirmed the presence of both Smc6 and Smc5 proteins in the peak, which is not immediately obvious from the gels in Figs 4B and 5C because of their very similar size and migration behavior. These results demonstrate that 4G assembly can be used not only to check the expression of complexes in a small-scale test, but also to purify homogeneous material from larger cultures.

## Discussion

We streamline the generation of expression construct variants for multisubunit protein complexes with new vectors (Table 1) and protocols. 4G cloning eliminates time-consuming steps such as fragment amplification and purification, and using sequence-verified donors, we create plasmids with up to four GECs in a single day in a sequence-conservative manner (Fig 1). Creation of user-specific ORF-donors is feasible in 3 d (day 1: PCR, Gibson assembly reaction, bacterial transformation; day 2: cloning picking; and day 3: miniprep and sequencing) once suitable primers are available. Overall, it is feasible to obtain multisubunit expression construct within 1 wk. The real advantage of the system becomes apparent once libraries of target ORF-donors (truncations, point mutations, etc.) have been created, because mutations can be quickly combined to create new versions of the target complex.

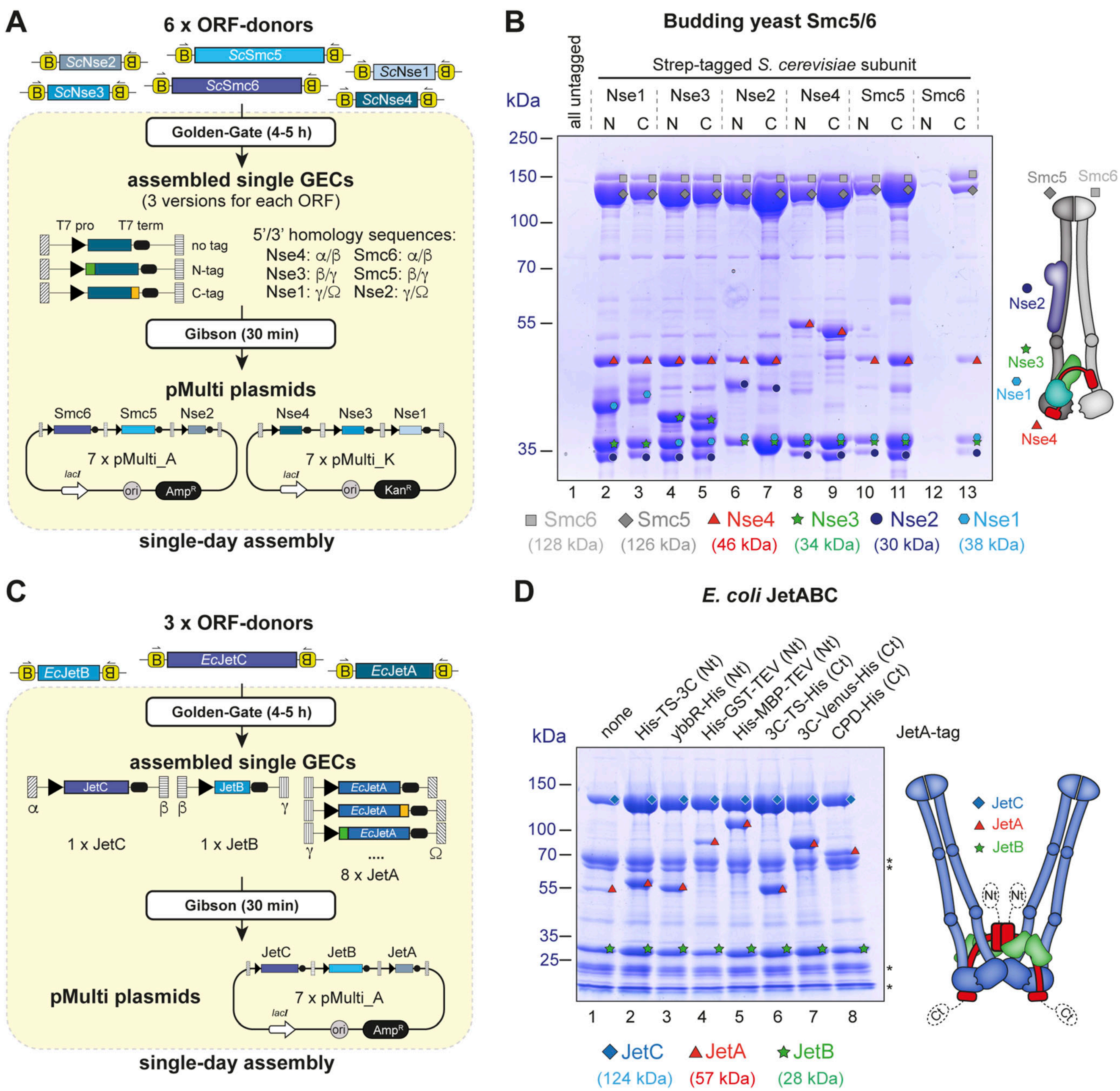


**Figure 3. Production and screening of prokaryotic expression vectors using 4G-cloning.**
**(A)** Overview of the cloning procedure to produce vectors for the expression of the *S. cerevisiae* Smc5/6 complex in *E. coli*. The six individual ORF-donors for Smc6, Smc5, Nse4, Nse3, Nse2, and Nse1 were used to create 18 GECs (each subunit either untagged, or N- or C-terminally tagged) containing T7 promoters and terminators, as well as dedicated Gibson overhangs to specify their position in the final vector. Subsequent Gibson assembly into pMulti_A or pMulti_K resulted in 14 vectors, which were used to obtain 13 variations regarding the placement of the affinity tag (see also Fig S2). **(B)** Coomassie-stained SDS gel showing the results of the expression screen for the *S. cerevisiae* Smc5/6 complex in *E. coli*. Only a C-terminal tag on Smc6 gives a stoichiometric complex after a single affinity step. The scheme on the right shows the overall architecture of the hexameric complex. Colored symbols for each subunit are used to identify the bands in the gel. **(C)** Overview of the cloning procedure to produce vectors for the expression of the *E. coli* JetABC complex. The three individual ORF-donors were used to create eight pMulti_A vectors differing in the tag added to the JetA subunit on the third position. **(D)** Coomassie-stained SDS gel showing the results of the expression screen for *E. coli* JetABC. Differences in yield can be observed when comparing different tags added to the JetA subunit. The untagged complex also has a certain affinity for the purification resin (lane 1). Additional bands represent contaminants and/or degradation products of subunits.
Source data are available for this figure.

Several points make 4G cloning unique and convenient: (i) "donors" for all GEC parts (promoters, tags, etc.) are sequence-verified only once, and can afterward be freely combined without further sequencing of products. (ii) One single Type IIS restriction enzyme (*BsaI*) is required to carry out all described cloning steps, minimizing sequence domestication efforts. (iii) Handling of DNA

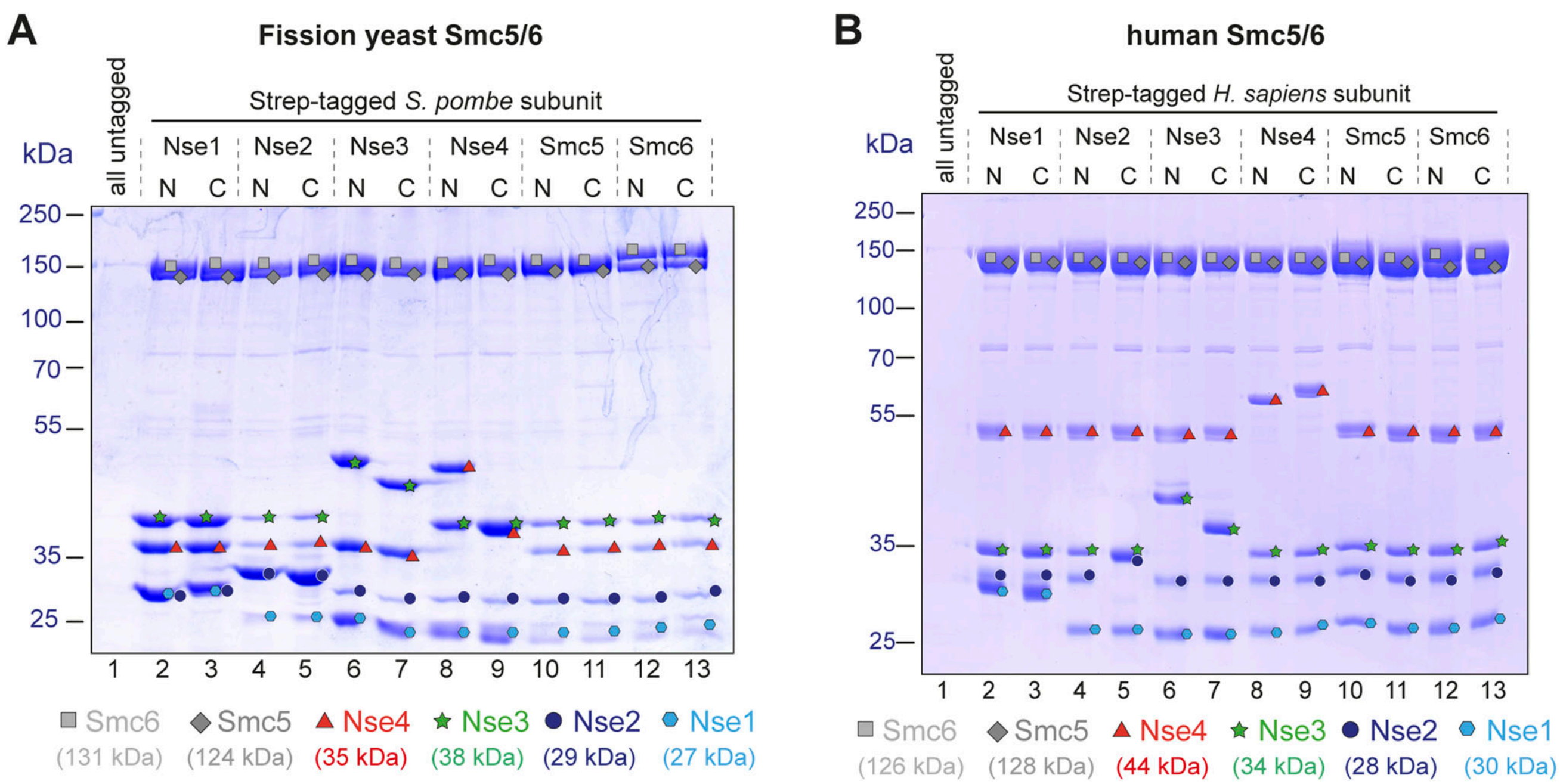


**Figure 4. Production and screening of eukaryotic expression vectors using 4G-cloning.**
**(A)** Coomassie-stained SDS gel showing the results of the expression screen for the *S. pombe* Smc5/6 complex in insect cells (*Sf9*) after infection with recombinant baculoviruses. For an overview of the cloning strategy for expression vectors, see Fig S3. Colored symbols for each subunit are used to identify the bands in the gel as in Fig 2B. Note that the apparent continuous downshift of Nse1 (and creation of a "double band") toward the right of the gel is caused by a problem with the gel at the bottom. **(B)** Coomassie-stained SDS gel showing the results of the expression screen for the *H. sapiens* Smc5/6 complex in mammalian cells. **(A)** Subunits are again identified with colored symbols as in (A).
Source data are available for this figure.

fragments is kept to a minimum without time-consuming purification steps. (iv) Cloning of expression plasmids containing multiple GECs can be done in a single day with minimal hands-on time. (v) Generation of many plasmid variants in parallel is straightforward and can be done with little additional effort.

The fact that important regulatory regions and protein-coding sequences are protected from assembly-induced mutations saves additional time and makes the procedure more cost-efficient. For each new project, only the new ORF-donors carrying individual subunits and not the final products for expression screening need to be sequenced. Taking our case for the budding yeast Smc5/6 complex as an example, this required sequencing of only 10.5 kbp for the six ORF-donors shown in Fig 3A and not the resulting >140 kbp of inserts from all 14 pMulti-plasmid versions. We have confirmed the sequence of selected expression constructs by whole-plasmid sequencing. Regulatory sequences and vector backbones are customizable, facilitating parallel construct creation for pro- and eukaryotic systems. If screening in several expression hosts is planned, it is, however, recommended to ensure that the codon usage of the used elements (e.g., ORFs) is suitable. Simultaneous transformation/transfection of several such multisubunit vectors allows for the expression of even larger assemblies (>10 subunits). For eukaryotic expression, we made our strategy compatible with biGBac vectors (3), allowing generation of even larger vectors containing up to 20 subunits, but it should be noted that this option requires sequence domestication of donors not only for *BsaI* but also for *PmeI* (3). With even larger complexes, our system is reaching a limit, and additional vectors and modifications to the protocol would become necessary. Regarding the rapid switching of expression hosts, it should be noted again that the ORF-donors might have to be adapted to meet the now host's codon requirements.

We validated our approach by screening various expression constructs for the hexameric Smc5/6 complex. Our results demonstrate that this eukaryotic assembly from yeasts and humans can be efficiently produced in *E. coli*, insect cells, or mammalian cells when all subunits are co-expressed. Our streamlined method enabled parallel generation of multiple plasmids in a single day for an unbiased screen to determine the optimal position of a specific affinity tag. Although tag positioning was less critical for *S. pombe* and human complexes in insect or mammalian cells (Fig 4), it was crucial for expressing the *S. cerevisiae* complex in *E. coli*, where only one option yielded a stoichiometric complex already after the affinity step. Once a suitable vector is cloned for WT complex expression, mutant versions of individual subunits can be created using the respective mutant donors. We extensively used this for the budding yeast complex to produce versions with ATPase and DNA-binding mutations or cysteine substitutions for chemical cross-linking (11, 13). In addition to screening for the best position for a given tag, several different tags for a given subunit can also easily be compared, and we applied this to the JetABC complex from *E. coli*.

Establishing 4G cloning in a new laboratory requires some investments, mainly in getting accustomed to the procedure and getting hold of the donor and acceptor vectors; however, any efforts are quickly offset by improved cloning and better material. Several projects in our laboratory have significantly benefited from 4G cloning.

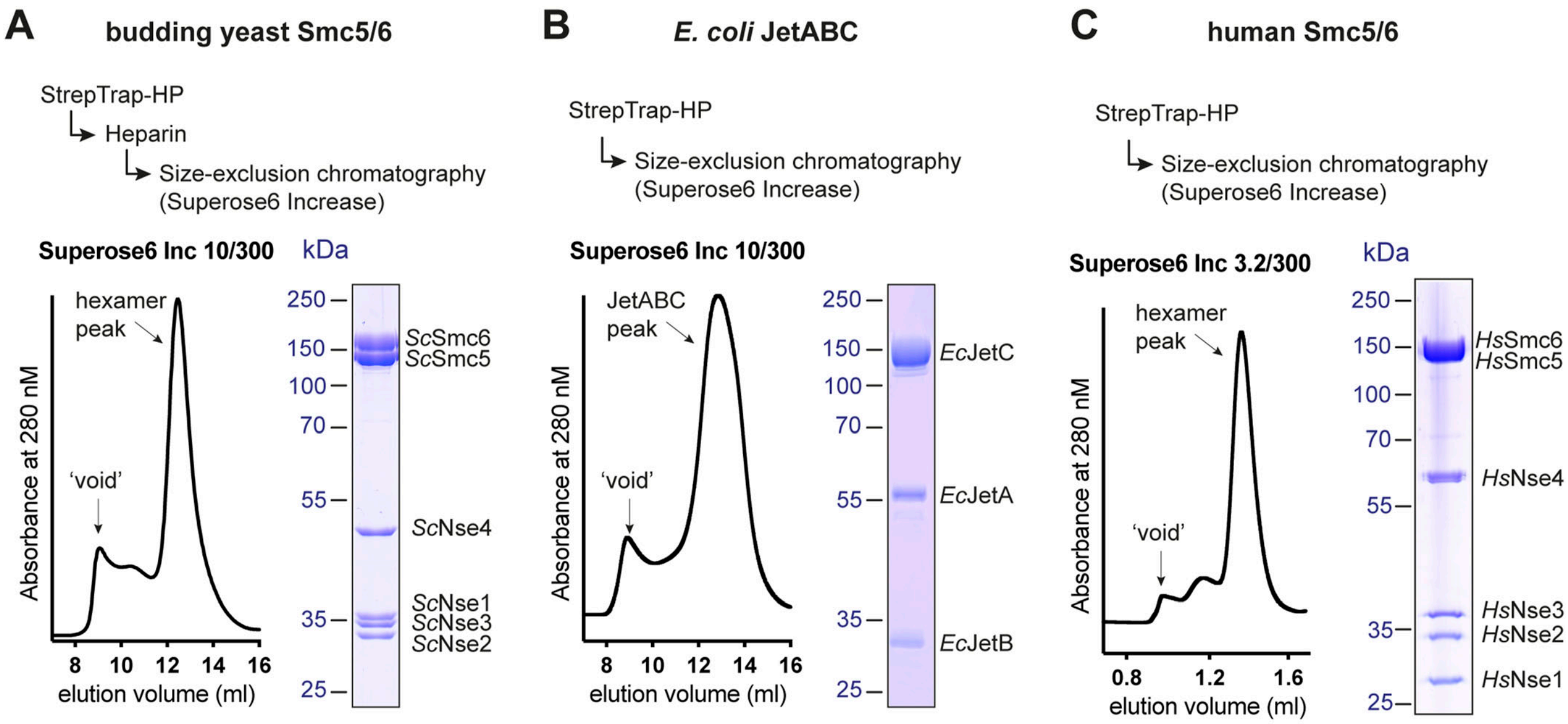


| | # peptides | % coverage |
|---|---|---|
| *Sc*Smc6 | 206 | 71 |
| *Sc*Smc5 | 163 | 67 |
| *Sc*Nse4 | 91 | 76 |
| *Sc*Nse3 | 69 | 69 |
| *Sc*Nse2 | 40 | 51 |
| *Sc*Nse1 | 44 | 78 |
| serum albumin | 4 | 7 |
| chloramphenicol transacetylase | 3 | 7 |

| | # peptides | % coverage |
|---|---|---|
| *Ec*JetC | 184 | 78 |
| *Ec*JetA | 63 | 74 |
| *Ec*JetB | 50 | 93 |
| serum albumin | 3 | 5 |
| thioredoxin | 1 | 9 |

| | # peptides | % coverage |
|---|---|---|
| *Hs*Smc6 | 214 | 72 |
| *Hs*Smc5 | 235 | 75 |
| *Hs*Nse4 | 101 | 79 |
| *Hs*Nse3 | 63 | 67 |
| *Hs*Nse2 | 39 | 77 |
| *Hs*Nse1 | 59 | 77 |
| histone H4 | 6 | 41 |
| histone H2B | 6 | 36 |

**Figure 5. Large scale production of selected targets.**
**(A, B, C)** Large-scale purifications of the *S. cerevisiae* Smc5/6 complex from 1 liter of *E. coli* culture (A), of the *E. coli* JetABC complex from 1 liter of *E. coli* culture (B), and of the human Smc5/6 complex from 1 liter of HEK293 culture (C). All panels show an outline of the purification steps (top), a chromatogram of the final size-exclusion chromatography step (middle, left), a Coomassie-stained gel showing the content of a fraction from the relevant peak (middle, right), and the top hits of a mass spectrometric analysis of the peak with a peptide count for each protein.
Source data are available for this figure.

## Materials and Methods

### Plasmids

The plasmids listed in Table 1 are available through Addgene, individually or as a kit (Addgene ID 1000000252). Donors for target ORFs need to be prepared by the user according to the described protocols. We highly recommend keeping stocks of all plasmids at the same molar concentration (we use 80 nM) as this makes it easier to mix them at an ideal ratio for GEC assembly. To roughly find the required stock concentration, we multiply the size of the plasmid in kb by 50, which gives the required concentration in ng/μl (e.g., 200 ng/μl for a 4-kb plasmid).

### Reagents

Gibson assembly mix is prepared as described in the original publication (9) with the exception that Taq DNA Ligase is not included in the mixture. For generating circular plasmids for transformation, ligation of the nicks is not required, and removing the ligase significantly reduces the cost per reaction. For Golden Gate assembly, *BsaI*-HF-v2 enzyme (NEB) and T4-DNA Ligase (1u/μl; Thermo Fisher Scientific) are used. Plasmids are maintained in the bacterial strains: DH5α (Thermo Fisher Scientific) for standard plasmids, pirHC (Geneva Biotech) for donor plasmids containing the R6Kγ replication origin, and *ccdB* Survival cells for empty pMulti-plasmids containing the *ccdB* cassette. For the expression of yeast proteins in *E. coli*, we use the *E. coli* Rosetta (DE3) strain (Novagen), and for the expression of the *E. coli* JetABC complex, we employ BL21(DE3)Gold. All bacterial cell lines are made chemically competent in-house following the standard procedures. Standard final antibiotic concentrations for growth of *E. coli* are 100 μg/ml for ampicillin, 50 μg/ml for kanamycin, 30 μg/ml for chloramphenicol, 10 μg/ml for tetracycline, and 7 μg/ml for gentamycin.

### Transformation of *E. coli* DH5α, BL21(DE3)Gold, and Rosetta(DE3)

Plasmid material for transformation is mixed with 50 μl of chemically competent *E. coli* cells and incubated on ice for 10 min in

1.5-ml Eppendorf tubes. The tubes are then heat-shocked by transferring them to a 42°C Eppendorf shaking incubator (without shaking) for 1 min. After a short (1–2 min) incubation on ice, 500 μl of LB medium is added and the cells are allowed to recover at 37°C in an Eppendorf shaking incubator (shaking at 700 rpm) for 45 min before plating on selective LB/agar plates. The plates are then incubated at 37°C overnight (around 16 h).

### Transformation of *E. coli* DH10 EMBacY

100 μl aliquots of chemically competent DH10 EMBacY cells are mixed with 2 μl of Bacmid DNA and incubated for 30 min on ice. The tubes are then heat-shocked by transferring them to a 42°C Eppendorf shaking incubator (without shaking) for 1 min. After a 10-min incubation on ice, 1 ml of LB medium is added and the cells are allowed to recover at 37°C in an Eppendorf shaking incubator (shaking at 700 rpm) for 4 h before plating on selective LB/agar plates (see the section of baculovirus generation). Here, we routinely prepare two plates per reaction, and we plate 5% of the material on one plate and 50% on the other to ensure that regardless of efficiency, we obtain a plate with a suitable number of well-separated colonies.

### Donor amplification for Gibson assembly

The donor vector (pD) is amplified and linearized for subsequent insertion of sequences by Gibson assembly by PCR using the primer pair pD_lin_fw (GGAGACCCACTGCTTGAGC) and pD_lin_rev (TGAGACCTAATATTCCGGAGTAG). 50 μl reactions contain 1x Phusion HF reaction buffer, 0.4 μM of each oligonucleotide, 0.4 mM dNTPs, 1 unit of Phusion DNA polymerase (NEB), and 10 pg of a pD-template. After an initial denaturation step at 95°C for 1 min, 30 PCR cycles with three steps (95°C for 10 s, 58°C for 10 s, and 72°C for 30 s) produce the desired product. A 5 μl aliquot is mixed with DNA loading dye and analyzed on a 1% agarose gel to verify amplification of the desired fragment. The product can be directly used for Gibson assembly, but it is recommended to purify it using a PCR purification kit (QIAGEN) to be able to accurately measure the DNA concentration. After determining it, the DNA is diluted to a final concentration of 25 ng/μl.

### Creation of donor vectors for ORFs and tags

Coding sequences for protein targets and for N- and C-terminal tags are amplified by PCR with primers containing appropriate overhangs for subsequent Gibson assembly into the linearized donor plasmid. For creating ORF-donors, these overhangs are 5′-GGAATATTA**GGTCTC**A<u>CCAT</u>G-3′ for the forward primer and 5′-CAAGCAGTG**GGTCTC**C<u>ATCC</u>-3′ for the reverse primer. *BsaI* sites that flank the insert and are used for Golden Gate assembly are shown in bold, the four underlined nucleotides show the specific "sticky end" created by *BsaI* digestion. In the forward primer, the three nucleotides on the 3′-end (ATG) anneal to the start codon for the ORF. When creating an ORF for an N-terminal truncation of a protein, an ATG has to be added as the first codon. The reverse primer has to be designed in a way to exclude the native stop codon if C-terminal tagging should be an option. Coding sequences for N-terminal tags are amplified with 5′-CTCCGGAATATTA**GGTCTC**A<u>GGCG</u>-3′ overhangs on forward primers and 5′-GGCTCAAGCAGTG**GGTCTC**C<u>ATGG</u>C-3′ overhangs on reverse primers, whereas for C-terminal tags, these overhangs are 5′-CTCCGGAATATTA**GGTCTC**A<u>GGAT</u>CC-3′ on forward and 5′-GGCTCAAGCAGTG**GGTCTC**C<u>CTTA</u>-3′ on reverse primers. If the sequence to be amplified does not contain internal *BsaI* sites, a single PCR using forward and reverse primers containing the overhangs described above is performed. In the case of internal sites, two additional primers and an additional PCR are needed for each site (see Fig S1C) to destroy these sites with silent mutations. PCRs contain the same components as described for linearization of the donor vector, and specific cycling conditions (annealing temperature, elongation time) are chosen for each target. A 5 μl aliquot of the PCR product is analyzed on an agarose gel to verify that the desired fragment was obtained and that it is free of non-specific amplification products or primer dimers, and to estimate product concentration. If only the desired product is visible, further purification is not necessary. Although this makes it impossible to accurately determine the concentration using spectrophotometric methods, we find it sufficient to estimate the concentration based on agarose gel electrophoresis. If the desired fragment is not the only amplification product, the whole PCR should be loaded on an agarose gel and the specific band purified using a gel extraction kit according to the manufacturer's instructions.

### Insertion of sequences into the linearized donor by Gibson assembly

1 μl of linearized donor vector (at 25 ng/μl) is mixed with PCR product(s) of the insert corresponding roughly to a 2–5 x molar excess of insert in PCR tubes. Sterile water is added to 5 μl final volume, and the tubes are put on ice. 5 μl of 2x Gibson-Mix is added on ice, and the tubes are quickly transferred to a preheated 50°C PCR block. After 15–30 min of incubation, the tubes are removed and placed on ice, and competent PIR1 *E. coli* cells (Invitrogen) are transformed with a 2 μl aliquot of the product. The bacteria are plated on two LB plates (for 90% or 10% of cells) containing 30 μg/ml chloramphenicol. After overnight incubation at 37°C, colonies are picked and grown in 3 ml LB liquid cultures containing 30 μg/ml chloramphenicol overnight (around 16 h) at 37°C. Cells from these cultures (OD[600 nm] typically between four and six) and plasmids are isolated using a miniprep kit (QIAGEN). Cloned inserts are sequenced using the primers pD_seq_fw (5′-CGCGGTACCATAACTTCGTATAGC-3′) and/or pD_seq_rev (5′-GGGGTTATGATAGTTATTGCTCAGCGG-3′). Internal primers in the inserts are also used for long sequences.

### Linearization of multi-GEC acceptors for Gibson assembly

pMulti_ccdB plasmids are linearized by *SwaI* digestion in 40 μl reactions containing 50 ng/μl (roughly 15 nM) plasmid DNA, 1x NEB buffer 3.1, and 10 U of *SwaI* enzyme. After incubation at 25°C for 2 h, 5 U of fresh enzyme is added followed by another 2 h at 25°C. The enzyme is then inactivated by incubation at 75°C for 20 min. The material can be directly used as Gibson-acceptor without further purification because the *ccdB* cassette avoids cloning background

because of an incompletely digested plasmid. Linearization of pBig1 plasmids for making expression constructs for insect and mammalian cells was done by *SwaI* digestion following a previously published protocol (3).

### Generation of multi-GEC plasmids by 4G cloning

To create GECs from donor plasmids by Golden Gate assembly, 10 μl reactions in PCR tubes are assembled containing 1x T4 DNA Ligase buffer (Thermo Fisher Scientific), 5 U of *BsaI*-HF-v2, 0.3 U of T4 DNA Ligase, and 8 nM (1 μl each of an 80-nM stock) of all donors required to produce the desired GEC (donors for ORF, N- or C-terminal tag, promoter/terminator, and 5′- and 3′-Gibson overhang). We recommend to prepare small aliquots of BsaI-HF-v2 and T4 DNA Ligase and to keep them frozen at −80°C, and to replace them if 4G cloning should fail.

Tubes are placed in a PCR machine and subjected to a cycling program (40 cycles of 37°C for 2 min and 16°C for 5 min, followed by a final step at 50°C for 10 min) to maximize the amount of fully assembled GECs. Compatible GECs created in this way are subsequently inserted by Gibson assembly into a suitable multi-GEC acceptor plasmid. For this, 2 μl of each GEC product is mixed with 1 μl of linearized multi-GEC acceptor (50 ng/μl) in PCR tubes on ice, the total volume is doubled using 2x Gibson-Mix, and tubes are quickly transferred to a PCR block preheated to 50°C. After incubation at this temperature for 30 min, DH5α competent cells are transformed with 5 μl of the product, and bacteria are plated on LB plates containing appropriate antibiotics and incubated at 37°C overnight.

### Expression tests in E. coli

Competent bacteria (*E. coli* Rosetta [DE3] for budding yeast Smc5/6, *E. coli* BL21(DE3)Gold for *E. coli* JetABC) are transformed using expression vectors and plated on LB plates supplemented with the appropriate antibiotics. Colonies are inoculated in 10 ml Terrific Broth (TB) with the same antibiotics and grown to an OD(600 nm) of about 1.0 at 37°C. The culture temperature is then decreased to 22°C, and expression is induced by adding 0.5 mM of IPTG followed by overnight incubation (around 16 h; final OD [600 nM] typically around 10) at this lower temperature. Cells are harvested by centrifugation (4,000*g*, 10 min), resuspended in lysis buffer (50 mM Tris–HCl, pH 7.5, 300 mM NaCl, 5% glycerol), and sonicated with an MS73 probe (40% output, 20 pulses of 1 s). The lysate is clarified by centrifugation at 16,000*g* at 4°C in a tabletop centrifuge, and the supernatant is added to Strep-Tactin Sepharose or $Ni^{2+}$-NTA Sepharose (both from Cytiva) affinity resin that has been washed and equilibrated in lysis buffer. Tubes are incubated at 4°C on a rotating wheel for 1 h, after which beads are pelleted by centrifugation (700*g*, 1 min) and washed twice with 1 ml lysis buffer. The bound material is eluted using 35 μl of lysis buffer supplemented with either 2.5 mM desthiobiotin (for Strep-Tactin) or 500 mM imidazole (for $Ni^{2+}$-NTA). After mixing the eluate with an equal volume of 2x SDS loading dye, the sample is boiled for 5 min and proteins are separated on a 12% SDS–PAGE gel followed by staining with Coomassie Brilliant Blue.

### Production of recombinant baculoviruses and protein expression tests in insect cells

DH10 EMBacY cells are transformed with vectors for insect cell expression (pBig) and plated on LB plates containing kanamycin (50 μg/ml), gentamycin (7 μg/ml), tetracycline (10 μg/ml), IPTG, and X-Gal. For IPTG and X-Gal addition, we spread 80 μl of IPTG (100 mM in water) and X-Gal (50 mg/ml in DMSO) stocks on top of plates containing antibiotics. After 36 h of incubation at 37°C, white colonies are picked and grown in 4 ml of LB containing kanamycin (50 μg/ml) and gentamycin (7 μg/ml). Cells are harvested (20,000*g*, 2 min, in an Eppendorf tabletop centrifuge) and resuspended in 300 μl of buffer P1 from the QIAGEN miniprep kit. 300 μl of buffer P2 from the QIAGEN miniprep kit is then added, and the mixture is incubated at room temperature for 5 min. After the addition of 300 μl of buffer N3 from the QIAGEN miniprep kit, the mixture is incubated in ice for another 5 min before cell debris is removed by centrifugation at 4°C in an Eppendorf tabletop centrifuge. 800 μl of the supernatant is mixed with 600 μl of isopropanol, and the tube is incubated at −20°C for 2 h to precipitate Bacmid DNA. After centrifugation for 10 min at 4°C at 20,000*g* in an Eppendorf tabletop centrifuge, the pellet is washed with 500 μl of 70% ethanol and spun again. The ethanol is removed, and the pellet is dried and resuspended in 50 μl of sterile water. 2 μl of this Bacmid DNA is transfected into *Sf9* cells in a plate of a six-well dish ($0.8 \times 10^6$ cells per well), and the cells are incubated at 28°C. As a control, we routinely include a well that is transfected with water. In this case, the cells overgrow during the subsequent incubation, whereas the cells in the wells transfected with Bacmid DNA stop dividing, visibly in size, and partially detach from the cell culture dish. After 4 d (96 h), the supernatants of these wells are harvested (500*g*, 10 min, in an Eppendorf tabletop centrifuge) and 200 μl is used to infect a 10 ml culture of *Sf9* cells at a density of $10^6$ cells/ml. After 3 d (72 h) of incubation at 28°C (shaking at 180 rpm), the cells are harvested (500*g*, 10 min in a centrifuge for Falcon tubes), and the pellets are processed as described for small-scale expression in *E. coli*.

### Expression tests in mammalian cell lines

Transfection of HEK293E cells with plasmid DNA (midi-prep scale) was carried out by the Protein Production and Structure Core Facility (PTPSP) at the Ecole Polytechnique Federale de Lausanne. Briefly, HEK293E cells are grown in suspension in Ex-Cell medium (Sigma-Aldrich) containing 4 mM glutamine. For 10 ml of expression tests, $10^7$ cells are harvested by centrifugation (450*g* for 6 min) and resuspended in RPMI 1640 medium (Invitrogen) containing 0.1% pluronic at a concentration of $20 \times 10^6$ cells/ml. 15 μg of plasmid DNA is added and mixed, followed by 30 μg of PEI-Max (Thermo Fisher Scientific). The mixture is then incubated for 90 min at 37°C with stirring, followed by dilution to $10^6$ cells/ml with Ex-Cell medium containing 4 mM glutamine and transfer to Spin Tubes. Valproic acid (VPA) is added to a final concentration of 3.75 mM, and the cultures are incubated for 3 d at 37°C. Cells are then harvested, and the pellets washed once with PBS and frozen in dry ice. For lysis, the pellets are thawed and resuspended in 2 ml of lysis buffer containing Benzonase and cOmplete protease inhibitors (Roche) and processed as described for small-scale expression in *E. coli*.

### Large-scale production of the Smc5/6 complex in *E. coli*

*E. coli* cells carrying the desired plasmids were grown in 1-liter cultures in TunAir flasks in TB medium at 37°C until they reached an OD(600 nm) of 1. The temperature was reduced to 22°C, expression was induced by the addition of 0.5 mM IPTG, and the culture was kept overnight shaking at 22°C. Cells were harvested by centrifugation (5,000*g*, 10 min), resuspended in 3–4 pellet volumes of lysis buffer (50 mM Tris–HCl, pH 7.5, 300 mM NaCl, 5% glycerol) supplemented freshly with 5 mM beta-mercaptoethanol and 1 mM PMSF, and sonicated with a VS70T probe (40% output, 15 min total time with ON/OFF cycles of 1 s each). The lysate was clarified by centrifugation at 40,000*g* for 30 min at 4°C, and the clear supernatant was then loaded onto a 5-ml StrepTrap column pre-equilibrated with lysis buffer (GE Healthcare). After washing the StrepTrap column with 5 column volumes (cV) of lysis buffer and 5 cV of Hep-A buffer (20 mM Tris–HCl, pH 7.5, 200 mM NaCl, 5% glycerol), a 5-ml HiTrap Heparin column (GE Healthcare) pre-equilibrated in Hep-A buffer was connected downstream of the StrepTrap column. The bound material was eluted from the StrepTrap column with Hep-A buffer supplemented with 2.5 mM desthiobiotin directly onto the Heparin column. After washing the Heparin column with 5 cV of Hep-A buffer, the bound material was eluted with a 5 cV gradient from Hep-A to Hep-B (20 mM Tris–HCl, pH 7.5, 1,000 mM NaCl, 5% glycerol). Peak fractions were analyzed by SDS–PAGE, and suitable fractions were concentrated to a final volume of 400 μl, and then loaded onto a Superose 6 Increase column equilibrated in SEC buffer (10 mM Tris–HCl, pH 7.5, 300 mM NaCl). Peak fractions were analyzed by SDS–PAGE.

### Large-scale production of the EcJetABC complex in *E. coli*

Cell growth, induction, lysis, and StrepTrap-HP chromatography were performed as described for purification of the yeast Smc5/6 complex in *E. coli*. After elution of the bound material with lysis buffer supplemented with 2.5 mM desthiobiotin, target-containing fractions were concentrated and loaded on a size-exclusion chromatography column (Superose 6 Increase 10/300 GL; Cytiva) equilibrated in SEC buffer (20 mM Tris–HCl, pH 7.5, 250 mM NaCl, 1 mM TCEP).

### Large-scale production of the human Smc5/6 complex in mammalian cells

Transfection of HEK293E cells with plasmid DNA (midi-prep scale) was carried out by the Protein Production and Structure Core Facility (PTPSP) at the Ecole Polytechnique Federal de Lausanne, following a procedure similar to the small-scale expression test but at the scale of a 1-liter culture. Cell pellets were harvested by centrifugation and frozen. The frozen pellet was resuspended in lysis buffer (50 mM Tris–HCl, pH 7.5, 300 mM NaCl, 5% glycerol) supplemented freshly with 7,000 U of Benzonase and 2 mM DTT, and then lysed by sonication with a VS70T probe (40% output, 5 min total time with ON/OFF cycles of 5 s each). The lysate was clarified by centrifugation at 40,000*g* for 30 min at 4°C, the supernatant was filtered through a 5-μM filter, and this material was then loaded onto a 5-ml StrepTrap column (GE Healthcare) pre-equilibrated with lysis buffer. The column was washed with 5 cV of lysis buffer, and the bound material was subsequently eluted with lysis buffer supplemented with 2.5 mM desthiobiotin. The eluate was concentrated, and a sample was analyzed by size-exclusion chromatography (Superose 6 Increase 3.2/300) to show that it contained a properly folded complex rather than a soluble aggregate.

### LC-MS/MS analyses

Samples were digested following a modified version of the iST method (23) (named miST method). Samples were heated for 10 min at 75°C and diluted 1:1 (v:v) with water. Reduced disulfides were alkylated by adding ¼ vol. of 160 mM chloroacetamide (32 mM final) and incubating for 45 min at RT in the dark. Samples were adjusted to 3 mM EDTA and digested with 1.0 μg trypsin/LysC mix (#V5073; Promega) for 1 h at 37°C, followed by a second 1-h digestion with an additional 1.0 μg of proteases. To remove sodium deoxycholate, two sample volumes of isopropanol containing 1% TFA were added to the digests, and the samples were desalted on a strong cation exchange (SCX) plate (Oasis MCX; Waters Corp.) by centrifugation. After washing with isopropanol/1% TFA, peptides were eluted in 150 μl of 40% MeCN, 19% water, and 1% (vol/vol) ammonia, and dried by centrifugal evaporation.

Tryptic peptide mixtures were injected on a Vanquish Neo nanoHPLC system interfaced via a Nanospray Flex source to a high-resolution Orbitrap Exploris 480 mass spectrometer (Thermo Fisher Scientific). Peptides were loaded onto a trapping microcolumn PepMap 100 C18 (5 mm × 1.0 mm ID, 5 μm, Thermo Fisher Scientific) before separation on a C18 custom–packed column (75 μm ID × 45 cm, 1.8-μm particles, ReproSil-Pur, Dr. Maisch), using a gradient from 2 to 80% acetonitrile in 0.1% formic acid for peptide separation at a flow rate of 250 nl/min (total time: 65 min). Full MS survey scans were performed at 120,000 resolution. A data-dependent acquisition method controlled by Xcalibur software (Thermo Fisher Scientific) was used that optimized the number of precursors selected ("top speed") of charge 2+ to 5+ while maintaining a fixed scan cycle of 2 s. Peptides were fragmented by higher energy collision dissociation (HCD) with a normalized energy of 30% at 15,000 resolution. The window for precursor isolation was 1.6 m/z U around the precursor, and selected fragments were excluded for 60s from further analysis.

MS/MS data were analyzed using Mascot 2.8 (Matrix Science) setup to search custom databases containing the specific yeast, human, and *E. coli* sequences, and the most usual environmental contaminants and enzymes used for digestion (keratins, trypsin, etc.). Trypsin (cleavage at K, R) was used as the enzyme definition, allowing two missed cleavages. Mascot was searched with a parent ion tolerance of 10 ppm and a fragment ion mass tolerance of 0.02 D. Carbamidomethylation of cysteine was specified in Mascot as a fixed modification. Protein N-terminal acetylation and methionine oxidation were specified as variable modifications. Scaffold (version Scaffold 5.3.3, Proteome Software Inc.) was used to validate MS/MS-based peptide and protein identifications. Peptide identifications were accepted if they could be established at greater than 95.0% probability by the Percolator posterior error probability calculation (24). Protein identifications were accepted if they could be established at greater than 99.0% probability and contained at

least five identified peptides. Protein probabilities were assigned by the Protein Prophet algorithm (25). Proteins that contained similar peptides and could not be differentiated based on MS/MS analysis alone were grouped to satisfy the principles of parsimony. Proteins identified by MS analysis representing cross-contamination between the three protein preparations were excluded from the summary tables in Fig 5. The file with the complete results of this analysis is available in Supplemental Data 1 and can be viewed with free Scaffold 5 software (https://www.proteomesoftware.com/products/scaffold-5).

## Data Availability

All raw data (gel and plate images and mass spectrometry output) are available as supplementary data.

## Supplementary Information


## Acknowledgements

We are grateful to members of the Gruber laboratory for stimulating discussions and advice during the development of 4G cloning, to Dr. Christian Biertümpfel for help with construct design, to the Protein Analysis Facility (PAF) at FBM-UNIL for protein identification by mass spectrometry, to the Protein Production and Structure Core Facility at EPFL for protein expression, and to Kelvin Lau for helpful feedback on the article. This work was supported by a grant from the Swiss National Science Foundation to S Gruber (170242).

### Author Contributions

M Taschner: conceptualization, data curation, formal analysis, validation, investigation, methodology, project administration, and writing—original draft, review, and editing.
JB Dickinson: investigation and writing—review and editing.
F Roisné-Hamelin: investigation and writing—review and editing.
S Gruber: conceptualization, supervision, funding acquisition, and writing—original draft, review, and editing.

### Conflict of Interest Statement

The authors declare that they have no conflict of interest.

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
