## [Reviewer comments · Life Science Alliance]

Life Science Alliance

4G cloning: rapid gene assembly for expression of multisubunit protein complexes in diverse hosts

Michael Taschner, Joe Dickinson, Florian Roisne-Hamelin, and Stephan Gruber

DOI: <https://doi.org/10.26508/lsa.202402899>

Corresponding author(s): *Stephan Gruber, University of Lausanne*

Review Timeline:

Submission Date:	2024-06-21
Editorial Decision:	2024-08-12
Revision Received:	2024-10-01
Editorial Decision:	2024-10-22
Revision Received:	2024-10-24
Accepted:	2024-10-25

Transaction Report:

August 12, 2024

Re: Life Science Alliance manuscript #LSA-2024-02899

Prof. Stephan Gruber
University of Lausanne
Department of Fundamental Microbiology
Quartier UNIL-Sorge
Batiment Biophore
Lausanne, Vaud 1015
Switzerland

Dear Dr. Gruber,

Thank you for submitting your manuscript entitled "4G cloning: rapid gene assembly for expression of multisubunit protein complexes in diverse hosts" to Life Science Alliance. The manuscript was assessed by expert reviewers, whose comments are appended to this letter. We invite you to submit a revised manuscript addressing the Reviewer comments.

Thank you for this interesting contribution to Life Science Alliance. We are looking forward to receiving your revised manuscript.

Sincerely,

B. MANUSCRIPT ORGANIZATION AND FORMATTING:

Reviewer #1 (Comments to the Authors (Required)):

The manuscript by Michael Taschner et al (LSA-2024-02899) reports the development of cloning strategy, termed 'Golden Gate-guided Gibson Assembly' that is termed 4G cloning that simplifies the assembly and revision of multigene expression cassettes.

In brief, the authors devised a cloning strategy based on Gibson assembly of Golden Gate-customized gene expression cassettes (GECs) which allows to produce final multi-gene expression vectors from sequence-verified elements in a single cloning step. They created an important set of building blocks, designated as "elements" that, provided the cDNA to be co-expressed have been properly domesticated, allow the creation of a multigene expression cassette in a single cloning step. As a proof of principle, the parallel construction of expression vectors for several Structural Maintenance of Chromosomes (SMC) complexes, which allowed to devise strategies for the recombinant production and optimization of these targets in bacteria, insect cells, and human cells is presented.

The proposed approach for assembling multigene expression constructs in diverse hosts is built on well-established cloning methods. The originality of the work lies in the design of a strategy that bypasses the need to verify single gene expression constructs before assembling them. If the investment required to efficiently implement 4G cloning turns out to be reasonable, the proposed single step procedure is likely to be widely used.

Overall, this methods manuscript is very interesting and proposes an appealing solution to streamline the assembly of multigene expression constructs. The document is well written but there is still room for improvements. Key direpoints are listed below:

Point 1: The assembly efficiency is precisely analysed for a limited set of constructs (Figure 2). The authors should provide estimates of the cloning efficiency experimented for the test cases presents in Figures 3 and 4.

Point 2: As generation of multi-GEC plasmids by 4G cloning is central to the manuscript, a detailed analysis of the parameters that influence the cloning efficiency would improve te manuscript.

Point 3: It could also be of interest to report failures and how they were rescued?

Reviewer #2 (Comments to the Authors (Required)):

4G cloning: rapid gene assembly for expression of multisubunit protein complexes in diverse hosts
Taschner, M., et al
Life Science Alliance

Taschner et al. report a novel way to generate expression vectors for multi-subunit protein complexes. The authors have created a cloning strategy that combines Gibson assembly and Golden Gate assemble, which they have called 'Golden Gate-guided Gibson Assembly' or "4G cloning". They propose that 4G cloning allows the creation many versions of expression plasmids in parallel (different tags, tag locations, and regulatory sequences), that can be expressed in different hosts with minimal alterations in the cloning process. The authors claim that this is quick and reliable assembly of many construct variations in parallel. The cloning technique was demonstrated by expressing the hexameric Smc5/6 complex from *Saccharomyces cerevisiae* into *Escherichia coli*, as well as Smc5/6 hexamer constructs for *Schizosaccharomyces pombe* and *Homo sapiens*, expressed in insect and mammalian cells, respectively.

The new cloning method reported is interesting and worth publishing in Life Science Alliance, yet major revisions are needed. Specifically, full application of the 4G cloning method needs to be shown. This would include a full protein purification and additional biophysical techniques that prove the protein complex has formed correctly. Overall, Coomassie stained expression gels do not correspond to correct protein complex assembly. Additionally, after the pulldown purification of each complex, a

Western blot or mass spectrum needs to be shown to indicate that the bands shown on the Coomassie stained gel is the corresponding protein. Finally, many details are lacking in the Materials and Methods sections and need to be expanded upon. There needs to be more transparency on how long each step of this method would take. Please see our more detailed comments below.

Recommended changes to the figures:

- Figure 2 - When transforming 4GECs into a vector, there was only a 40% success rate, which is not very high. Yet, in the introduction the authors discuss that "For multi-subunit complexes the individual subunits can be expressed from separate vectors which are co-delivered into host cells, but this becomes inconvenient and unreliable for larger assemblies (>3 subunits). A better alternative is to produce multiple proteins from the same vector, with each subunit being expressed from its own gene expression cassette (GEC) with appropriate regulatory sequences"
- Figure 2 - Was this experiment only performed once? To get a more accurate percentage, this should be repeated
 - o Was any software used to quantify the number of colonies on the plate? Using a program to quantify the plate could yield more accurate results.
- We recommend the addition of a figure or expand upon Figures 3 and 4: Add Western blots for all Coomassie stained gels with the appropriate antibodies to conclude which band corresponds to which each protein of interest in each complex. This could be further supported and strengthened with Mass Spectrometry experiments
 - o Figure 4 - Two proteins run at the same molecular weight. How do you know both proteins are being expressed? For example, the bands for Smc5 and Smc6 were merged into the pulldown using C terminal tagged Smc5.
- Figure 3B and 3D - Why are the molecular weights of the subunits different in pulldown with different subunit tags. For example, Nse4 subunit has a higher molecular weight in pulldown using the tagged Nse4 than pulldown with other subunit tags.
- Figure 3B - Add an explanation and interpretation for the results lane 12

Overall recommended changes to the manuscript:

- This is an innovative cloning approach, but it may be limited only to the screening of the tag positions or tags used. The use of these constructs to purify the complex is not demonstrated. Was the Smc5/6 complex or three-subunit Wadjet SMC complex or any other complex purified using this strategy?
- Authors need to discuss the limitations of the methods in the discussion
- On page 7, in reference to Figure 3, the authors state "Pulldowns created high amounts of material." Specify and quantify what a high amount of material means.
- It is not clear how long this whole protocol would take. The authors claim that the first steps take 4-5 hours. However, in the methods it is stated that "The product can be directly used for Gibson assembly, but it is recommended to purify it using a PCR purification kit (Qiagen) to be able to accurately measure the DNA concentration". If this step is recommended, I feel the total time for this process is misleading. Additionally, the cloning would take more time if internal Bsal sites needed to be removed from protein.
- Currently the vectors used for 4G cloning are not available from Addgene. What will the cost be for these vectors? The authors claim this is a more cost-effective cloning method, but that is difficult to determine without more information.

Recommended changes to the Materials and Methods:

- The authors used the Rosetta strain of E.coli for the yeast proteins and BL21(DE3)Gold cells for JetABC
 - o Why were these cell lines chosen? Why are they different for the two constructs? There was no discussion about the selection of cell lines. This could drastically change the expression results for this paper. (Page 11)
 - o In the section 'Insertion of sequences into linearized donor by Gibson assembly', how are the 3mL cultures grown? What media? For how long? Which antibiotics? How many hours was it grown? What were the final ODs?
- More details are needed for the protein expression protocol in E.coli:
 - o How many hours were the cells grown overnight? What were the final ODs?
 - o How many replicates were performed?
 - o What concentration of antibiotics were used?
 - o Give more details on how the cells were harvested (rpm? Time? Temperature?)
 - o How long is the lysate centrifuged for after sonication?
 - o After the high-speed spin, is the supernatant being added to the resin? It was not clear.
 - o How was the resin washed / equilibrated?
- More details are needed for the protein expression protocol in insect cells (sf9s):
 - o What is the concentration of antibiotic and X-Gal in the plates?
 - o How were the cells grown? At what temperature? How many hours?
 - o When purifying a Bacmid, usually you use a DNA purification kit, what kit did you use? The protocol only states the use of Buffer P1 and P2?
 - o Were any controls run for the transduction?
- In the methods section, more details are needed for the transformation protocol. The flow chart in Figure 1 says the transformation protocol takes 1 hour, but most protocols from companies that make the competent cell take 1.5-2 hours.
 - o Which antibiotics were used and at which concentrations?
 - o What temperature were the plates grown at?
 - o Were the plates grown overnight? How many hours?
 - o What concentration of antibiotics do you use in the plates?
 - o What were the controls for the transformations?
 - o What volume of cells per plated?

Reviewer #3 (Comments to the Authors (Required)):

In the manuscript by Taschner et al. entitled '4G Cloning: Rapid Gene Assembly for Expression of Multisubunit Protein Complexes in Diverse Hosts', the authors present a recombinant production system for large multisubunit protein assemblies amenable to bacterial and eukaryotic hosts. Their novel concept combines two well established cloning strategies that rely on Golden Gate cloning and Gibson assembly. Their Golden Gate-guided Gibson Assembly (4G cloning) approach is based on a collection of plasmids that can be combined in a flexible, modular, and fast (one day) manner. The authors benchmarked their recombinant protein production system using bacterial and eukaryotic multiprotein assemblies that belong to the structural maintenance of chromosomes (SMC) complex family. The authors compellingly demonstrate the feasibility and timesaving aspects of their system for producing variants of the SMC complexes from *Saccharomyces cerevisiae*, *Schizosaccharomyces pombe*, and *Homo sapiens*, as well as the bacterial JetABC complex. The methodology is designed to be versatile and efficient, reducing the time and effort required to produce complex gene assemblies. I support the publication of their results in Life Science Alliance with some experimental revisions and clarifications in the main text and figures as outlined below.

Remarks/comments:

1. The authors should mention and discuss that additional N- or C-terminal residues are added to the natural polypeptide sequence of the produced proteins. As 4G cloning leaves a minimal seam of only a single extra amino acid to either side of the protein, neither authentic N and C termini of the proteins of interest are maintained, nor the presence of potential N-terminal acetylation sites. For instance, presence of natural termini was important to produce recombinant versions of the anaphase promoting complex/cyclosome (APC/C) for structural studies (Zhang et al. 2016 Methods; PMID: 26454197 and references therein).
2. The authors explain conclusively that their cloning system can be used to switch expression hosts easily. However, they do not comment on how codon usage in the various expression hosts impacts protein production when switching hosts without adapting the coding sequences. A direct comparison should be easily realizable by exchanging promoters and terminators using their 4G cloning strategy, e.g. by directly comparing expression yields of one of the SMC complexes in *E. coli* vs. mammalian cells.
3. The authors present pulldown experiments using affinity chromatography matrixes, such as StrepTactin or Ni²⁺-NTA Sepharose, to show successful production of the protein complexes of interest. The authors claim that placements of affinity tags, at either N or C termini of different subunits, impacts subunit stoichiometries and yields. Although the SDS-PAGE analyses hint towards differences in stoichiometries of the individual complexes, their monodispersities and activities should be shown by additional approaches, e.g. size exclusion chromatography and/or activity assays. Furthermore, the authors claim on page 7 (Figure 3B) that 'When the tag was placed at the N-terminus of Smc6 the amount of protein obtained in the pulldowns was minimal'. The authors should convince the reader that the low yield really depends on the inaccessibility of the tag or on low expression levels, e.g. by immunoblotting using whole cell extracts. Furthermore, the authors do not state how the individual subunits indicated in each gel were identified? How often were the pulldowns repeated to show reproducibility?
4. It should be explained better why the authors chose to use a His6-tag to isolate JetABC, when they knew that the JetC subunit binds to Ni²⁺-NTA matrices. The authors should also revisit the last sentence on page 7 'Among those some clearly produced more protein than others, with the previously described construct (Fig. 3D, lane 2) being among the options with superior yield.' and render the statements more precise. What do the authors refer to when they state 'Among those some...'? What means 'more protein' in terms of amounts (mg, µg...)? What is superior yield?
5. The authors present a well-balanced collection of donor vectors that encode contemporary protease-cleavable and non-cleavable tags as listed in Table 1. However, the exact amino acid sequences of the designed tags are not amenable to the reader. To allow an educated decision-making which combination of tags are best for a certain biological question, the authors should provide in a separate table amino acid sequences of their tag designs (entire sequences of large commonly used tags, such as maltose-binding protein (MBP), for instance, could be omitted).
6. Although the authors state that sequencing of final expression vectors after 4G cloning is not necessary, they should mention that all final vectors generated in their study were sequenced by whole plasmid sequencing already in the beginning of the results part and not only in the discussion.
7. The authors state on page 4 (bottom) that they 'successfully removed up to five internal BsaI sites in a single step in this way'. In which constructs were these domestications made and why is this important for their study? The sentence 'The remaining set of vectors (Table 1) will be made available from AddGene.' should be rephrased once the manuscript is accepted.
8. Furthermore, the authors claim that 'all donors have a backbone of 1833 bp'. That is a trivial statement, since they apparently used an identical plasmid backbone. They should rather state that all donors originate from the same parental plasmid, thus all resulting pD vectors confer resistance to chloramphenicol and carry the R6ky origin of replication. The size ranges -from smallest to largest donor- might be more useful to the reader/potential user. The authors should furthermore specify if their 'acceptor vectors for the final multi-GEC assemblies allowing for expression in bacteria' also contain compatible replicons (e.g. Duet vectors, Novagen) for stable co-existence in bacterial hosts.
9. Despite being more streamlined than other cloning strategies for multisubunit complex production, the process presented by Taschner et al. still requires considerable amount of reagents and maintenance. The authors should provide a more realistic estimation of the time frame of the entire process from initial ORF cloning to protein production than just dwelling on the generation of multi-GEC assemblies within one day.
10. The authors should cite and discuss recent literature describing different cloning and protein production strategies, e.g. the

USER system or genomic tagging approaches.

11. Some phrases in the manuscript remain very vague. For example, on page 6 the authors state that 'similar vectors for a different host can be created very quickly'. They should precise what 'very quickly' means.

Minor formatting issues:

- I have noticed that the authors use the term transformation in a wrong way; not the plasmids are transformed but the bacterial cells are. Please correct.
- The term restriction digest is colloquial and should be rephrased.
- A prime symbol should be used to indicate ends of nucleic acid sequences, e.g. 5 prime and 3 prime ends, and no apostrophe.
- 'Gel filtration' should read 'size exclusion chromatography'.
- Please cite the correct paper for biGBac vectors (page 9; Weissman).
- Restriction enzymes should be written in italics; the same is true for Sf9 cells.
- Specify 'Rosetta strain' on page 11.
- Is the 5'- CAAGCAGTGGGTCTCCATCC-3' a reverse primer?
- Page 13, bottom: 700 grams or 700 xg?
- Page 14: final concentrations of antibiotics should be provided.
- Figure 1: For instance, 'X' and 'Y', as well as 'B', are not defined in the figure caption. Please proofread and extend the figure captions accordingly.
- Figure 2, caption: Which competent strains were used?
- Figure 3: Marker lanes should be indicated properly with a dash. Calculated molecular weights should be provided for each subunit. Was the sample lost in lane 12? Why do proteins with the same N- and C-terminal tags migrate differently, e.g. upshift of Nse1 in lanes 2/3, downshift of Nse2 in lanes 8/9 (also compare to downshift of Nse1 in Fig. 4, lanes 2/3)? Why is there a double band for Nse1 in Fig. 3, lane 5?
- Figure 4: Marker lanes should be indicated properly.
- Supp. Figure 1: Dashed lines in panel C?
- Suppl. Figure 2: Misplace arrow below ScSmc6.

We thank all three reviewers for their careful examination of the manuscript and their support and constructive criticism. We have made modifications to the manuscript as detailed below.

Reviewer #1 (Comments to the Authors (Required)):

The manuscript by Michael Taschner et al (LSA-2024-02899) reports the development of cloning strategy, termed 'Golden Gate-guided Gibson Assembly' that is termed 4G cloning that simplifies the assembly and revision of multigene expression cassettes.

In brief, the authors devised a cloning strategy based on Gibson assembly of Golden Gate-customized gene expression cassettes (GECs) which allows to produce final multi-gene expression vectors from sequence-verified elements in a single cloning step. They created an important set of building blocks, designated as "elements" that, provided the cDNA to be co-expressed have been properly domesticated, allow the creation of a multigene expression cassette in a single cloning step. As a proof of principle, the parallel construction of expression vectors for several Structural Maintenance of Chromosomes (SMC) complexes, which allowed to devise strategies for the recombinant production and optimization of these targets in bacteria, insect cells, and human cells is presented.

The proposed approach for assembling multigene expression constructs in diverse hosts is built on well-established cloning methods. The originality of the work lies in the design of a strategy that bypasses the need to verify single gene expression constructs before assembling them. If the investment required to efficiently implement 4G cloning turns out to be reasonable, the proposed single step procedure is likely to be widely used.

Overall, this methods manuscript is very interesting and proposes an appealing solution to streamline the assembly of multigene expression constructs. The document is well written but there is still room for improvements. Key direpoints are listed below:

Point 1: The assembly efficiency is precisely analysed for a limited set of constructs (Figure 2). The authors should provide estimates of the cloning efficiency experimented for the test cases presents in Figures 3 and 4.

We have now included statistics on cloning efficiencies in a new Figure 2 with bar charts regarding colony numbers and percentage of positive clones. We also removed the '1 GEC' case, because it is not a useful case (4G cloning is not necessary to create such a simple plasmid). While we did not quantify the cloning efficiency when producing the constructs for the test cases shown in Figures 3 and 4, in all cases we got a plate with plenty of colonies, screened two clones from each plate, and typically this gave at least one positive hit. These constructs had 3 GECs, with one of them having a tag in most cases, and thus it is slightly more 'complex' than the '3 GEC' situation in the presented test case and could thus be somewhat less efficient.

We decided against repeating the cloning for all the expression constructs to determine efficiencies as this number will inevitably be slightly different for each lab carrying out a similar experiment. In our case we make our own Gibson-Assembly mix and also make our bacteria chemically competent to keep the costs low. Most likely, when using commercial reagents (especially commercially available ultra-competent cells) even better numbers can be obtained.

We now made modifications to the text to highlight the fact that these numbers are merely shown to convince the reader that assembly of 4 GECs into a vector is easily feasible.

Point 2: As generation of multi-GEC plasmids by 4G cloning is central to the manuscript, a detailed analysis of the parameters that influence the cloning efficiency would improve te manuscript.

We have not systematically looked at improving the assembly conditions by varying parameters, as the proper clones are normally obtained with the standard procedure. We noticed however that it is important to mix the starting plasmids at stoichiometric amounts and to follow the proper cycling protocol for the GoldenGate part of the reaction in order to maximise the formation of properly assembled GECs. All this info is now highlighted in the Materials and Methods section.

Point 3: It could also be of interest to report failures and how they were rescued?

In cases where assembly failed, it was usually rescued by either starting with freshly preped plasmid DNA (likely because DNase contaminations inevitably inhibit GEC formation), or by getting a new tube of enzyme (BsaIHF-v2 or T4 DNA Ligase) from the freezer. Regarding enzymes, we realised that 4G cloning seems quite sensitive to enzyme activity, thus we store small aliquots of enzymes at -70 °C. This information has now been added to the Methods section.

Reviewer #2 (Comments to the Authors (Required)):

4G cloning: rapid gene assembly for expression of multisubunit protein complexes in diverse hosts
Taschner, M., et al
Life Science Alliance

Taschner et al. report a novel way to generate expression vectors for multi-subunit protein complexes. The authors have created a cloning strategy that combines Gibson assembly and Golden Gate assembly, which they have called 'Golden Gate-guided Gibson Assembly' or "4G cloning". They propose that 4G cloning allows the creation of many versions of expression plasmids in parallel (different tags, tag locations, and regulatory sequences), that can be expressed in different hosts with minimal alterations in the cloning process. The authors claim that this is a quick and reliable assembly of many construct variations in parallel. The cloning technique was demonstrated by expressing the hexameric Smc5/6 complex from *Saccharomyces cerevisiae* into *Escherichia coli*, as well as Smc5/6 hexamer constructs for *Schizosaccharomyces pombe* and *Homo sapiens*, expressed in insect and mammalian cells, respectively.

The new cloning method reported is interesting and worth publishing in Life Science Alliance, yet major revisions are needed. Specifically, full application of the 4G cloning method needs to be shown. This would include a full protein purification and additional biophysical techniques that prove the protein complex has formed correctly. Overall, Coomassie stained expression gels do not correspond to correct protein complex assembly. Additionally, after the pulldown purification of each complex, a Western blot or mass spectrum needs to be shown to indicate that the bands shown on the Coomassie stained gel are the corresponding protein. Finally, many details are lacking in the Materials and Methods sections and need to be expanded upon. There needs to be more transparency on how long each step of this method would take. Please see our more detailed comments below.

Recommended changes to the figures:

- Figure 2 - When transforming 4GECs into a vector, there was only a 40% success rate, which is not very high. Yet, in the introduction the authors discuss that "For multi-subunit complexes the individual subunits can be expressed from separate vectors which are co-delivered into host cells, but this becomes inconvenient and unreliable for larger assemblies (>3 subunits). A better alternative is to produce multiple proteins from the same vector, with each subunit being expressed from its own gene expression cassette (GEC) with appropriate regulatory sequences"

Considering that four cassettes are here assembled directly from individual elements into an expression vector (a total of 21 DNA fragments) we do not consider 40% to be a bad efficiency. With increasing complexity due to increasing number of elements it is expected that errors occur leading to the exclusion of individual cassettes. With a chance of 40% it simply means that instead of mini-prepping a single colony one would have to prep 3 or 4 and analyse it by restriction digest. Doing mini-preps for 4 clones instead of one or two hardly increases the workload.

- Figure 2 - Was this experiment only performed once? To get a more accurate percentage, this should be repeated
- o Was any software used to quantify the number of colonies on the plate? Using a program to quantify the plate could yield more accurate results.

Replicate data are now included and shown in the new Fig. 2B. We removed the '1 GEC' case because it wouldn't really make sense to clone a vector for expression of a single protein by 4G assembly, and because the colony numbers were always so high that it would have been hard to see the difference between the '4 GECs' sample and the control in the bar graph. For colony counting we used the VisionWorks software (AnalytikJena), and manually inspected/corrected the colony count. This information has been added to the manuscript.

- We recommend the addition of a figure or expand upon Figures 3 and 4: Add Western blots for all Coomassie stained gels with the appropriate antibodies to conclude which band corresponds to which each protein of interest in each complex. This could be further supported and strengthened with Mass Spectrometry experiments

While most subunits can be uniquely identified from the protein gels, we have now performed mass spec analysis on selected samples (Figure 4). As expected, the results show that all expected subunits are present and that little additional proteins (contaminants) are detectable. Moreover, for each case we include a control containing a fully untagged complex, which does not yield material (with the exception of the JetABC complex due to its known 'stickiness' to Ni-NTA resin). Furthermore, the fact that individual subunits change their mobility when they are tagged leaves no doubt that we see these subunits there, and not some unrelated contaminant which would not show altered mobility. In addition, we have now analysed the purified preparations of proteins by proteomics confirming the presence of all subunits and the absence of major contaminations.

- o Figure 4 - Two proteins run at the same molecular weight. How do you know both proteins are being expressed? For example, the bands for Smc5 and Smc6 were merged into the pulldown using C terminal tagged Smc5.

In multiple lanes in Fig. 4A one can actually see that there are 2 bands for Smc5 and Smc6 when zooming in, especially in those with slightly less protein (e.g. lanes 2, 3, 4, 7). In the lanes with tagged Smc6 it is obvious. In Fig. 4B it is less obvious due to higher protein amounts and thicker bands, but with tagged Smc6 again it is clearest. For the human complex we now purified the version with a C-terminal tag on Nse4 to homogeneity (new Figure 5C). Here it is not at all obvious from the small-scale test that both proteins are there, but we show by mass-spectrometric analysis that it is the case.

- Figure 3B and 3D - Why are the molecular weights of the subunits different in pulldown with different subunit tags. For example, Nse4 subunit has a higher molecular weight in pulldown using the tagged Nse4 than pulldown with other subunit tags.

(see also response to reviewer 3) The addition of a purification tag changes the molecular weight of the respective protein (and its charge) and leads to an altered (usually reduced) electrophoretic mobility. The direction of change in mobility and magnitude however not always predictable. Apparently, the electrophoretic mobility of individual subunits changes not only with molecular weight and charge, but also with charge distribution. Thus, even proteins with (almost) identical amino acid composition may exhibit distinct mobility (see Nse4 N and C samples in Fig. 3B).

- Figure 3B - Add an explanation and interpretation for the results lane 12

This explanation is already provided in the following sentence: "When the tag was placed at the N-terminus of Smc6 the amount of protein obtained in the pulldowns was minimal (lane 12), indicating that Smc6 does not tolerate this tag position. ". Some proteins can simply not be tagged/purified using tags at certain positions, and this seems to be the case for N-terminal tags (at least the 3C-TwinStrep tag used here) on budding yeast Smc6 after expression in *E. coli*. Whether this is due to inhibition of expression of this subunit or inaccessibility of the tag is currently unknown but given that the C-terminal tag gives good results we did not investigate further. This result strongly underscores the need to test multiple tag and tag positions highlighting the usefulness of 4G cloning.

Overall recommended changes to the manuscript:

- This is an innovative cloning approach, but it may be limited only to the screening of the tag positions or tags used. The use of these constructs to purify the complex is not demonstrated. Was the Smc5/6 complex or three-subunit Wadjet SMC complex or any other complex purified using this strategy?

It is true that this small-scale expression screening for multi-subunit expression construct doesn't necessarily mean that the complex can be purified to homogeneity by large-scale expression, and it is possible that the obtained material will be aggregated. We have now included data that demonstrate that 3 of the 4 cases produce well behaved protein as judged by size exclusion chromatography (budding yeast Smc5/6 and the JetABC complex in bacteria, and human Smc5/6 in mammalian cells) (new Fig. 5). For the JetABC complex we chose the N-terminal His-TwinStrep-3C tag on JetA, but given the 'stickiness' of the complex on Ni-NTA resin we used the TwinStrep component for purification, which eliminated the problem. In all test cases we obtained several mg of soluble material as shown by the final size-exclusion chromatography step. We would also like to point out that we used constructs prepared by 4G cloning extensively for previous publications, which we reference in the manuscript. We purified >20 versions of the budding yeast Smc5/6 complex and used them for characterization of its DNA-dependent ATPase, cysteine-crosslinking of protein interfaces, and topological entrapment assays of double-stranded DNA (Taschner et al, EMBO J 2021; Taschner and Gruber, NSMB 2023. For the JetABC complex we used the material for analysis of circular plasmid restriction and structural investigations by cryo-electron-microscopy (Liu et al, Mol Cell 2022; Roisné-Hamelin et al, Mol Cell 2024). We therefore have no doubt that these complexes are properly folded and functional.

- Authors need to discuss the limitations of the methods in the discussion

We have now added that expression of complexes with >10 subunits in bacteria and of complexes with >20 subunits in eukaryotic hosts becomes inconvenient with our current set of vectors and protocols, and that further modifications would be required to make this work. We also pointed out in the discussion (as well as in other parts of the manuscript) that switching of expression hosts might necessitate codon-optimization of donor sequences for optimal expression results.

- On page 7, in reference to Figure 3, the authors state "Pulldowns created high amounts of material." Specify and quantify what a high amount of material means.

We agree that the use of the word 'high' was not appropriate and have changed this now to 'amounts of material easily detected by Coomassie-staining', which is usually easily sufficient to move forward to large-scale expression. This is the main purpose of such small-scale test.

- It is not clear how long this whole protocol would take. The authors claim that the first steps take 4-5 hours.

However, in the methods it is stated that "The product can be directly used for Gibson assembly, but it is recommended to purify it using a PCR purification kit (Qiagen) to be able to accurately measure the DNA concentration". If this step is recommended, I feel the total time for this process is misleading. Additionally, the cloning would take more time if internal BsaI sites needed to be removed from protein.

The sentence the reviewer refers to relates to the production of the individual ORF-donor vectors (and not the final 4G-cloning procedure outlined in Figure 1). Here the target ORF is amplified using primers containing homology overhangs to the pD donor vector. If such a PCR yields a single band we typically use it directly for the Gibson Assembly reaction, thus saving time. In this case it is, however, not possible to accurately measure the DNA concentration and thus one cannot mix linearized pD and the PCR product at optimal (equimolar) ratios. Despite this we always obtain the desired product. Adding the recommended PCR purification step (e.g. Qiagen PCR purification kit) to be able to determine the concentration adds only about 10 minutes to this step. In general, it is of course true that while the 4G assembly step itself can be done in a single day, the entire process from ORF-donor preparation to mini-prepping of the final expression construct is more time-consuming. We now add a paragraph in the discussion with an estimation of the time-frame for the entire process. We also mentioned that the real advantage of this cloning procedure becomes apparent once a good tagging/purification strategy for the wildtype complex is created, and extensive libraries of mutant ORF-donors have been created. Combining mutations in different subunits with each other is then straightforward, and we have used this extensively to clone/purify >20 versions of the budding yeast Smc5/6 complex (as referenced in the discussion).

- Currently the vectors used for 4G cloning are not available from Addgene. What will the cost be for these vectors? The authors claim this is a more cost-effective cloning method, but that is difficult to determine without more information.

It is true that ordering individual vectors from AddGene can become costly with increasing number of plasmids. However, AddGene has kindly offered to supply all 49 plasmids listed in Table 1 as a kit. These kits are typically sold for around 400 USD. Given that this purchase has to be made only once, the time saved for cloning and screening of expression constructs over the course of many years to come justifies this investment.

Recommended changes to the Materials and Methods:

- The authors used the Rosetta strain of *E. coli* for the yeast proteins and BL21(DE3)Gold cells for JetABC
 - o Why were these cell lines chosen? Why are they different for the two constructs? There was no discussion about the selection of cell lines. This could drastically change the expression results for this paper. (Page 11)

For the budding yeast Smc5/6 complex we directly amplified the ORFs from the yeast genome (as indicated in the Results section), and these sequences were thus not codon-optimized for bacteria. This is the reason why we chose the Rosetta strain as it contains tRNA genes for rare codons in bacteria. The JetABC complex is from *E. coli* and thus the standard bacterial codons present in BL21(DE3)Gold cells is appropriate in this case. We mention these details now in the manuscript.

- o In the section 'Insertion of sequences into linearized donor by Gibson assembly', how are the 3mL cultures grown? What media? For how long? Which antibiotics? How many hours was it grown? What were the final ODs?

The cultures are standard miniprep cultures. They are grown in LB medium containing 30 µg/ml of Chloramphenicol (given the Chloramphenicol resistance gene in the pD vector). They were grown overnight (around 16 h) to a final OD of roughly 4-6, but we do not routinely measure ODs for the purpose of mini-preps and hence didn't consider these details essential. We included this information in the revised methods section.

- More details are needed for the protein expression protocol in *E. coli*:
 - o How many hours were the cells grown overnight? What were the final ODs?

After induction of protein expression by IPTG of the cultures grown in TB medium the cells were grown at the lower temperature for about 16 h. The final ODs in these cases are routinely around 10. This is now included in the methods section.

- o How many replicates were performed?

For these expression tests we did not perform replicates. But we went ahead and purified some examples in large-scale based on the results of the small-scale tests and could obtain homogeneous material of the respective complexes (now included in the manuscript).

- o What concentration of antibiotics were used?

Standard antibiotic concentrations are 100 µg/ml for Ampicillin, 50 µg/ml for Kanamycin, 30 µg/ml for Chloramphenicol, 7 µg/ml for Gentamycin, and 10 µg/ml for tetracycline. These details are now added the methods section (Reagents).

o Give more details on how the cells were harvested (rpm? Time? Temperature?)

This is now included

o How long is the lysate centrifuged for after sonication?

This has now been added to the appropriate method sections.

o After the high-speed spin, is the supernatant being added to the resin? It was not clear.

Yes, this is usually how soluble proteins are purified and this has now been added to the appropriate method section

o How was the resin washed / equilibrated?

Both equilibration and wash of the resin is done with Lysis buffer. This has now been added to the appropriate method sections.

• More details are needed for the protein expression protocol in insect cells (sf9s):

o What is the concentration of antibiotic and X-Gal in the plates?

added

o How were the cells grown? At what temperature? How many hours?

We believe that this information was already there. 28°, for either 3 or 4 days, shaking at 180 rpm in the case of the 10 ml culture. We now added that 3 days corresponds to 72 h, and 4 days corresponds to 96 h.

o When purifying a Bacmid, usually you use a DNA purification kit, what kit did you use? The protocol only states the use of Buffer P1 and P2?

We use buffers P1, P2, and N3 from the Qiagen mini-prep kit, but prepare the Bacmid DNA by precipitation rather than using a column due to its large size. We now specify that the buffers are from the Qiagen kit and left the rest of the description as it was before because it gives a detailed step-by-step protocol.

o Were any controls run for the transduction?

We routinely use a control which we treat only with the transfection reagent but without Baculovirus DNA. The cells in this well overgrow during the 4 days of incubation, whereas the cells in the wells with DNA stop growing and show signs of virus production (larger cells, detachment from the plate, etc.). This information has now been added.

• In the methods section, more details are needed for the transformation protocol. The flow chart in Figure 1 says the transformation protocol takes 1 hour, but most protocols from companies that make the competent cell take 1.5-2 hours.

1 hour is enough for our bacterial transformation procedure, the details of which have now been added to the methods section.

o Which antibiotics were used and at which concentrations?

Details on antibiotic concentrations have been added. See also the response to an earlier question.

o What temperature were the plates grown at?

Plates with E. coli are always grown at 37°C. This information has now been added.

o Were the plates grown overnight? How many hours?

Cells were grown for 36 h until the colonies were either blue or white, and large enough to be picked for inoculation of 3 ml mini-prep cultures. This information has now been added.

o What concentration of antibiotics do you use in the plates?

Details on antibiotic concentrations have been added. See also the response to an earlier question.

o What were the controls for the transformations?

We do not routinely include controls for such transformations. Only after preparing a new batch of competent DH10 EMBacY cells we ensure by plating on plates with IPTG and X-gal that all colonies are blue. After transfection of Bacmid DNA we then select only white colonies.

o What volume of cells per plated?

This information has now been added.

Reviewer #3 (Comments to the Authors (Required)):

In the manuscript by Taschner et al. entitled '4G Cloning: Rapid Gene Assembly for Expression of Multisubunit Protein Complexes in Diverse Hosts', the authors present a recombinant production system for large multisubunit protein assemblies amenable to bacterial and eukaryotic hosts. Their novel concept combines two well established cloning strategies that rely on Golden Gate cloning and Gibson assembly. Their Golden Gate-guided Gibson Assembly (4G cloning) approach is based on a collection of plasmids that can be combined in a flexible, modular, and fast (one day) manner. The authors benchmarked their recombinant protein production system using bacterial and eukaryotic multiprotein assemblies that belong to the structural maintenance of chromosomes (SMC) complex family. The authors compellingly demonstrate the feasibility and timesaving aspects of their system for producing variants of the SMC complexes from *Saccharomyces cerevisiae*, *Schizosaccharomyces pombe*, and *Homo sapiens*, as well as the bacterial JetABC complex. The methodology is designed to be versatile and efficient, reducing the time and effort required to produce complex gene assemblies. I support the publication of their results in Life Science Alliance with some experimental revisions and clarifications in the main text and figures as outlined below.

Remarks/comments:

1. The authors should mention and discuss that additional N- or C-terminal residues are added to the natural polypeptide sequence of the produced proteins. As 4G cloning leaves a minimal seam of only a single extra amino acid to either side of the protein, neither authentic N and C termini of the proteins of interest are maintained, nor the presence of potential N-terminal acetylation sites. For instance, presence of natural termini was important to produce recombinant versions of the anaphase promoting complex/cyclosome (APC/C) for structural studies (Zhang et al. 2016 Methods; PMID: 26454197 and references therein).

We would like to point out that the N-termini of ORFs do not contain any additional amino acids and are native (unless of course an N-terminal tag is added). The C-termini of ORFs will have an additional glycine residues in the absence of a C-terminal tag. If a native C-terminus is needed for a particular protein, the native STOP codon can be included during cloning of the ORF-donor, and the fully native untagged protein can be produced this way. This of course eliminates the option of C-terminal tagging using this construct. This information has now been added to the Results section.

2. The authors explain conclusively that their cloning system can be used to switch expression hosts easily. However, they do not comment on how codon usage in the various expression hosts impacts protein production when switching hosts without adapting the coding sequences. A direct comparison should be easily realizable by exchanging promoters and terminators using their 4G cloning strategy, e.g. by directly comparing expression yields of one of the SMC complexes in *E. coli* vs. mammalian cells.

Many thanks. This is a valid point that we have overlooked in the original ms. Simple re-cloning with other promoters/terminators will not change the codons of the ORF. When such ORF sequences are synthesized one could choose codons which are more or less suitable for several hosts to overcome this problem. We now point this out in the manuscript.

3. The authors present pulldown experiments using affinity chromatography matrixes, such as StrepTactin or Ni²⁺-NTA Sepharose, to show successful production of the protein complexes of interest. The authors claim that placements of affinity tags, at either N or C termini of different subunits, impacts subunit stoichiometries and yields. Although the SDS-PAGE analyses hint towards differences in stoichiometries of the individual complexes, their monodispersities and activities should be shown by additional approaches, e.g. size exclusion chromatography and/or activity assays. Furthermore, the authors claim on page 7 (Figure 3B) that 'When the tag was placed at the N-terminus of Smc6 the amount of protein obtained in the pulldowns was minimal'. The authors should convince the reader that the low yield really depends on the inaccessibility of the tag or on low expression levels, e.g. by immunoblotting using whole cell extracts. Furthermore, the authors do not state how the individual subunits indicated in each gel were identified? How often were the pulldowns repeated to show reproducibility?

We now add a new Figure 5 where we show the purification of three test examples to homogeneity and show that they are properly folded. We only did this for constructs which we concluded to be superior based on small-scale expression tests. We also had mentioned in the manuscript that we already used complexes cloned by 4G assembly for several studies and also references these studies. For the *S. cerevisiae* hexameric Smc5/6 complex for example we analysed regulation of the ATPase activity (Taschner et al, EMBO J 2021) and topological DNA loading (Taschner and Gruber, NSMB 2023). For the JetABC complex we used the purified material for Electron Microscopy and plasmid restriction assays (Liu et al, Mol Cell 2022; Roisné-Hamelin et al, Mol Cell 2024). Regarding the absence of pulldown material in the case of a C-terminal Smc6 tag we do not know if this is caused by lack of expression or tag inaccessibility, and we now point this out. However, given the success of the C-terminal Smc6-tag and the resulting purification of properly assembled hexamer (new Figure 5A) we do not care, because the screen yielded a good construct. This case strongly underscores the need to screen tags and tag positions.

4. It should be explained better why the authors chose to use a His6-tag to isolate JetABC, when they knew that the JetC subunit binds to Ni²⁺-NTA matrices. The authors should also revisit the last sentence on page 7 'Among those some clearly produced more protein than others, with the previously described construct (Fig. 3D, lane 2) being among the options with superior yield.' and render the statements more precise. What do the authors refer to when they state 'Among those some...'? What means 'more protein' in terms of amounts (mg, µg...)? What is superior yield?

Here we wanted to directly compare different tags on either the N- or C-terminus of JetA on the obtained amounts of pulldown material. To do this we chose tag-combinations which all contained a His-component, so we could use the same resin for direct comparison. It is true that the protein purity after His purification is poor, but we now show in the new Figure 5B that using the same construct (His-TwinStrep-3C-JetA / JetC / JetB) that simply using the Twin-Strep component for purification overcomes the 'stickiness' and yields properly assembled complex. We now clarify that 'more protein' is based on judging band intensities in the gel. We didn't attempt to quantify protein amounts (in µg) from these small-scale tests. We also removed the word 'superior yield' because we agree with the reviewer that this was a poor choice of wording.

5. The authors present a well-balanced collection of donor vectors that encode contemporary protease-cleavable and non-cleavable tags as listed in Table 1. However, the exact amino acid sequences of the designed tags are not amenable to the reader. To allow an educated decision-making which combination of tags are best for a certain biological question, the authors should provide in a separate table amino acid sequences of their tag designs (entire sequences of large commonly used tags, such as maltose-binding protein (MBP), for instance, could be omitted).

This is a good point. However, rather than including a table full of printed amino-acid sequences which is complicated to analyze, we prefer that the readers download the plasmid sequences/maps for the donors directly from AddGene where they are all available (see Table 1 for accession codes).

6. Although the authors state that sequencing of final expression vectors after 4G cloning is not necessary, they should mention that all final vectors generated in their study were sequenced by whole plasmid sequencing already in the beginning of the results part and not only in the discussion.

We did whole plasmid sequencing on a small selection of final plasmids (as mentioned) just to confirm that there are no errors. As suggested, we state this now already in the results section. The user is of course free to sequence any or all of the assembled expression constructs.

7. The authors state on page 4 (bottom) that they 'successfully removed up to five internal BsaI sites in a single step in this way'. In which constructs were these domestications made and why is this important for their study?

This was in a construct that was not used in this study. To avoid confusion, we now removed this sentence. We just wanted to point out that even if one target sequence has several internal BsaI sites, the donor vector can still be created in a single step by simultaneous introduction of many silent mutations in the Gibson-Assembly reaction. The reason why these domestications are important (even essential) for our study is that subsequent assembly of GECs is done by BsaI-shuffling, and any internal site for this restriction enzyme would chop up the GEC and prevent assembly of the desired vector.

The sentence 'The remaining set of vectors (Table 1) will be made available from AddGene.' should be rephrased once the manuscript is accepted.

Yes, we are still waiting for the ordering number of the complete plasmid kit and will include this information upon acceptance.

8. Furthermore, the authors claim that 'all donors have a backbone of 1833 bp'. That is a trivial statement, since they apparently used an identical plasmid backbone. They should rather state that all donors originate from the same parental plasmid, thus all resulting pD vectors confer resistance to chloramphenicol and carry the R6ky origin of replication. The size ranges -from smallest to largest donor- might be more useful to the reader/potential user. The authors should furthermore specify if their 'acceptor vectors for the final multi-GEC assemblies allowing for expression in bacteria' also contain compatible replicons (e.g. Duet vectors, Novagen) for stable co-existence in bacterial hosts.

We removed this piece of information now. The reason why we initially wanted to emphasize this is that this vector can be very efficiently linearized by PCR due its small size.

9. Despite being more streamlined than other cloning strategies for multisubunit complex production, the process

presented by Taschner et al. still requires considerable amount of reagents and maintenance. The authors should provide a more realistic estimation of the time frame of the entire process from initial ORF cloning to protein production than just dwelling on the generation of multi-GEC assemblies within one day.

We added an updated time-frame in the discussion which includes also the time required to clone and sequence-verify the user-specific ORF donors. Overall, it will take 3 days to create these donors (not including the time it takes to order the required primers and wait for their delivery), and the entire procedure of cloning a multi-subunit expression construct is feasible within one week.

10. The authors should cite and discuss recent literature describing different cloning and protein production strategies, e.g. the USER system or genomic tagging approaches.

We initially considered mentioning other cloning procedures, but then decided against it because it became too confusing (too many different cloning procedures). We were not aware of the USER system but given that it entirely relies on PCR we don't see how this one would be relevant to discuss. Genomic tagging is of course an alternative option to purify complexes, but it is not at all compatible for 'quick' screening of different tags and tag-positions. To keep the manuscript concise, we prefer to leave it as it is.

11. Some phrases in the manuscript remain very vague. For example, on page 6 the authors state that 'similar vectors for a different host can be created very quickly'. They should precise what 'very quickly' means.

We agree that this was not a good choice. We removed the word 'quickly' now.

Minor formatting issues:

- I have noticed that the authors use the term transformation in a wrong way; not the plasmids are transformed but the bacterial cells are. Please correct.

Corrected.

- The term restriction digest is colloquial and should be rephrased.

We now replaced this term with 'EcoRV digest'.

- A prime symbol should be used to indicate ends of nucleic acid sequences, e.g. 5 prime and 3 prime ends, and no apostrophe.

Corrected

- 'Gel filtration' should read 'size exclusion chromatography'.

Corrected

- Please cite the correct paper for biGBac vectors (page 9; Weissman).

Thanks. This has escaped our attention and has now been corrected.

- Restriction enzymes should be written in italics; the same is true for Sf9 cells.

Corrected

- Specify 'Rosetta strain' on page 11.

Done

- Is the 5'- CAAGCAGTGGGTCTCCATCC-3' a reverse primer?

Yes, this has now been indicated.

- Page 13, bottom: 700 grams or 700 xg?

Thanks, corrected.

- Page 14: final concentrations of antibiotics should be provided.

All antibiotic concentrations have now been added.

- Figure 1: For instance, 'X' and 'Y', as well as 'B', are not defined in the figure caption. Please proofread and extend the figure captions accordingly.

This has now been added as a legend in Figure 1 below the schematic representation of the 'pD' donor vector

- Figure 2, caption: Which competent strains were used?

This information has now been added

- Figure 3: Marker lanes should be indicated properly with a dash. Calculated molecular weights should be provided for each subunit. Was the sample lost in lane 12? Why do proteins with the same N- and C-terminal tags migrate differently, e.g. upshift of Nse1 in lanes 2/3, downshift of Nse2 in lanes 8/9 (also compare to downshift of Nse1 in Fig. 4, lanes 2/3)? Why is there a double band for Nse1 in Fig. 3, lane 5?

Markers have now been corrected and expected MW for subunits (untagged!) have now been added. Lane 12 contains minimal amount of pulldown material because tagging of Smc6 at the N-terminus does not yield complex for pulldown. Whether this is due to lack of expression or inaccessibility of the tag is currently unknown. However, we didn't follow up on this because the screen clearly showed that C-terminal tagging of Smc6 gives stoichiometric complex, which was confirmed by large scale purification of the complex now shown in the new Figure 5.

Nse1 shifts up in lanes 2/3 because in these lanes we load complex which contains a tag on Nse1, thus adding molecular weight to this protein. Nse2 in lanes 8/9 does not shift down, the protein runs like in all other lanes in which this subunit is not tagged. Nse2 is only tagged in lanes 6/7, where is also clearly shifts up due to higher MW caused by addition of tags.

The double-band in Fig 3 lane 5 is not Nse1 but Nse3. In this case it contains a C-terminal tag, and possibly the tag sequence got slightly degraded here

In Figure 4 lanes 2/3 Nse1 runs higher because it is tagged. In the remaining wells it is untagged and thus runs lower. The further continuous 'downshift' towards the right (and the creation of a 'double band' towards the very right of the gel) is caused by a problem with the gel. This has now been indicated in the Figure legend.

- Figure 4: Marker lanes should be indicated properly.

Corrected

- Supp. Figure 1: Dashed lines in panel C?

Lines have now been removed

- Suppl. Figure 2: Misplace arrow below ScSmc6.

Thank you, now corrected

October 22, 2024

RE: Life Science Alliance Manuscript #LSA-2024-02899R

Prof. Stephan Gruber
University of Lausanne
Department of Fundamental Microbiology
Quartier UNIL-Sorge
Batiment Biophore
Lausanne, Vaud 1015
Switzerland

Dear Dr. Gruber,

Thank you for submitting your revised manuscript entitled "4G cloning: rapid gene assembly for expression of multisubunit protein complexes in diverse hosts". We would be happy to publish your paper in Life Science Alliance pending final revisions necessary to meet our formatting guidelines.

- please address Reviewer 2's remaining comment
- please be sure that the authorship listing and order is correct
- please add the Twitter handle of your host institute/organization as well as your own or/and one of the authors in our system
- please add the author contributions to the main manuscript text
- please consult our manuscript preparation guidelines <https://www.life-science-alliance.org/manuscript-prep> and make sure your manuscript sections are in the correct order
- please add a panel A to your Figure 2 legend

A. FINAL FILES:

B. MANUSCRIPT ORGANIZATION AND FORMATTING:

Sincerely,

Reviewer #2 (Comments to the Authors (Required)):

The revised version meets my expectations. Please include the mass spectrometry spectra as a supplementary figure.

Reviewer #3 (Comments to the Authors (Required)):

The authors of the manuscript entitled '4G cloning: rapid gene assembly for expression of multisubunit protein complexes in diverse hosts' answered all my points raised during the review process in a satisfactory manner. I therefore support publishing in Life Science Alliance.

October 25, 2024

RE: Life Science Alliance Manuscript #LSA-2024-02899RR

Prof. Stephan Gruber
University of Lausanne
Department of Fundamental Microbiology
Quartier UNIL-Sorge
Batiment Biophore
Lausanne, Vaud 1015
Switzerland

Dear Dr. Gruber,

Thank you for submitting your Methods entitled "4G cloning: rapid gene assembly for expression of multisubunit protein complexes in diverse hosts". It is a pleasure to let you know that your manuscript is now accepted for publication in Life Science Alliance. Congratulations on this interesting work.

DISTRIBUTION OF MATERIALS:

Again, congratulations on a very nice paper. I hope you found the review process to be constructive and are pleased with how the manuscript was handled editorially. We look forward to future exciting submissions from your lab.

Sincerely,
